# From Mice to Trains:
# Amortized Bayesian Inference on Graph Data

**Svenja Jedhoff**
*Department of Statistics, TU Dortmund University, Dortmund, Germany*

**Elizaveta Semenova**
*School of Public Health, Imperial College London, London, UK*

**Aura Raulo**
*Department of Biology, University of Oxford, Oxford, UK*

**Anne Meyer**
*Department of Mechanical Engineering, Karlsruhe Institute of Technology, Karlsruhe, Germany*

**Paul-Christian Bürkner**
*Department of Statistics, TU Dortmund University, Dortmund, Germany*

**Reviewed on OpenReview:** *https://openreview.net/forum?id=vpIeCm7YEA*

## Abstract

Graphs arise across diverse domains, from biology and chemistry to social and information networks, as well as in transportation and logistics. Inference on graph-structured data requires methods that are permutation-invariant, scalable across varying sizes and sparsities, and capable of capturing complex long-range dependencies, making posterior estimation on graph parameters particularly challenging. Amortized Bayesian Inference (ABI) is a simulation-based framework that employs generative neural networks to enable fast, likelihood-free posterior inference. We adapt ABI to graph data to address these challenges to perform inference on node-, edge-, and graph-level parameters. Our approach couples permutation-invariant graph encoders with flexible neural posterior estimators in a two-module pipeline: a summary network maps attributed graphs to fixed-length representations, and an inference network approximates the posterior over parameters. In this setting, several neural architectures can serve as the summary network. In this work we evaluate multiple architectures and assess their performance on controlled synthetic settings and two real-world domains — biology and logistics — in terms of recovery and calibration.

## 1 Introduction

Fast and accurate estimation of statistical quantities remains a central challenge in modern statistics. Beyond conventional data modalities such as tabular data and time series, many problems are most naturally expressed as graph-structured data — collections of vertices and edges $G = (V, E)$ that encode relational structure. Graphs arise when the outcome of interest depends on relations rather than independent observations. Examples for this are molecular graphs and protein-protein interaction networks in biology and chemistry (Besharatifard & Vafaee, 2024; Pfeifer et al., 2022; Gilmer et al., 2017), but also transportation, supply and power networks in operation and infrastructure (Jiang & Luo, 2022; Li et al., 2024; Wasi et al., 2024; Gharaee et al., 2021; Pagani & Aiello, 2013) can be naturally structured as graphs. Other examples are social and information networks and interaction graphs in agent-based systems (Sharma et al., 2024; Eden et al., 2021; Battaglia et al., 2016; Chopra et al., 2021). Typical inference targets range from node-level latents to edge parameters and global graph-level parameters, with the added complication that both the

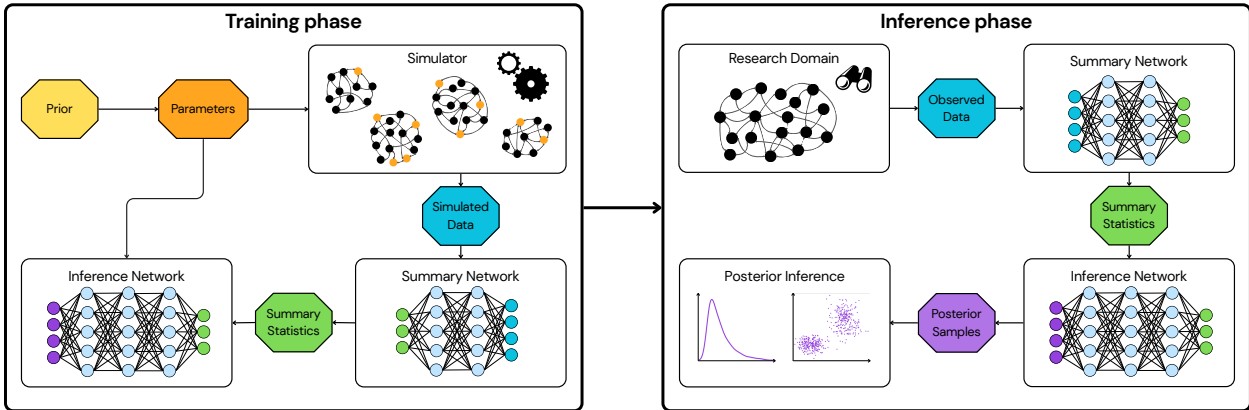

Figure 1: Overview of the proposed framework, which comprises two phases. In the training phase (left), parameters are sampled from the prior and passed to a simulator to generate graph-structured datasets. These simulated graphs are then used to jointly train the summary network (encoder) and the inference network (posterior estimator). In the inference phase (right), an observed graph is processed by the trained summary and inference networks to obtain near instant posterior inference on the parameters.

topology and the attributes may be random (Newman, 2010). In our experiments, we consider applications in biology and logistics, where graph-level parameters are of interest.

Inference for parameters on graph-structured data poses several unique challenges. First, permutation symmetry implies that models, likelihoods, and posteriors should be invariant to node relabeling (Bronstein et al., 2021). Otherwise, arbitrary labeling induces artificial multi-modality. Graph isomorphism means that the same unlabeled graph can be represented by multiple vertex-edge combinations and automorphism can create additional non-identifiabilities if not handled carefully (Bollobás, 1998). Second, graphs vary in order and density: both $|V|$ and $|E|$ differ across instances, degree distributions are often heavy-tailed, and sparsity leads to highly imbalanced neighborhoods — complicating batching, memory usage, and statistical efficiency (Chiang et al., 2019). Third, long-range dependencies are hard to capture: local message passing tends to over-smooth features, which makes distant nodes look alike (Cai & Wang, 2020). Simply adding more layers in the neural models rarely fixes this and can hurt optimization and uncertainty calibration.

Amortized Bayesian Inference (ABI) (Gershman & Goodman, 2014; Papamakarios & Murray, 2018; Radev et al., 2023b; Ritchie et al., 2016; Gloeckler et al., 2024) provides a compelling solution to a series of problems like repeated posterior inference in simulator-based models, real-time updating in sequential decision-making, and scalable analyses across large collections of related datasets: By jointly training multiple neural networks to approximate posterior distributions, ABI enables rapid and repeated inference even when the likelihood is intractable. ABI underpins several strands of simulation-based inference, including neural likelihood estimation and neural posterior estimation (Papamakarios et al., 2019; Mittal et al., 2025; Radev et al., 2023a). Despite notable successes in various fields like uncertainty aware surrogate modeling (Scheurer et al., 2025), Bayesian mixture models (Pakman & Paninski, 2018; Kucharský & Bürkner, 2025), Bayesian multilevel models (Habermann et al., 2025; Arruda et al., 2025; 2024; Wu et al., 2024) and experimental design (Huang et al., 2024; Kennamer et al., 2023), existing ABI pipelines are not tailored to graphs and face new challenges when working with graph-structured data. The limited prior work in this direction either focuses on spatial data (Sainsbury-Dale et al., 2025) or addresses narrow, problem-specific settings (Stillman et al., 2023), leaving a general framework for graph-structured ABI largely undeveloped.

We adopt a two-module pipeline (see Figure 1), as defined by Radev et al. (2023b): a summary (encoder) network that maps a graph $G = (V, E)$ with attributes to a fixed-length representation $h(x)$, and an inference network that approximates the conditional posterior $p(\theta \mid h(x))$ over parameters $\theta$. Since $h(x)$ is generally not sufficient, the learned posterior is conditioned on an information-reduced representation of the data. Consequently, the accuracy and calibration of $p(\theta \mid h(x))$ are bounded by how much parameter-relevant information the encoder preserves, making the summary network the critical design choice. When handling

graph data instead of standard tabular data, the summary network must (i) respect permutation invariance so that node relabeling does not change $h(x)$; (ii) handle graphs of variable size and sparsity; and (iii) capture long-range dependencies. Because we train with simulated data, the simulation process must cover different families of graphs in terms of size and sparsity patterns observed in practice. Otherwise, a distributional mismatch between simulated and real graphs will increase the amortization gap (Cremer et al., 2018), causing the amortized posterior to become systematically biased and poorly calibrated on real data.

In this paper we introduce a graph-aware ABI framework that couples permutation-invariant neural network encoders with flexible posterior estimators to enable fast, scalable inference on graphs. Figure 1 provides an overview of the framework: the left panel illustrates the training phase, where the summary and inference networks are trained on simulated data, and the right panel shows the inference phase, where the trained model is used to perform posterior inference for an observed graph. We instantiate several deep architectures that balance receptive field and efficiency, and analyze their performance with the help of multiple case studies. Our evaluation spans a controlled synthetic setting and two real-world domains — biology and logistics — where we target node-, edge-, and graph-level parameters.

## 2 Methods

We develop a method for efficient ABI on graph-structured data. We begin by formalizing the problem and describing the relevant graph structures, followed by a brief introduction to ABI. Our approach employs a summary network and an inference network; because the summary network must be tailored to graph-structured inputs, this work focuses primarily on its design. Accordingly, we present a series of deep neural architectures suitable as summary networks. In Section 2.4, we introduce metrics used to compare configurations, and in Section 3, we evaluate these setups empirically.

### 2.1 Problem setting

In our Bayesian formulation, we model parameters $\theta$ and observed data $D$ jointly via $p(\theta, D) = p(\theta)\, p(D \mid \theta)$. Here, $D$ denotes an observed graph, and the latent parameters $\theta$ govern the structural and statistical properties of this graph.

A graph is defined by its set of $N$ vertices $V = \{1, \ldots, N\}$ together with edges $E \subseteq V \times V$. The connectivity can be represented by an $N \times N$ adjacency matrix $A$, where (for an unweighted graph) $a_{ij} = 1$ indicates an edge between vertices $i$ and $j$, and $a_{ij} = 0$ otherwise. In addition, each vertex may carry attributes; we collect $p$ attributes per vertex in a matrix $X \in \mathbb{R}^{N \times p}$, where row $i$ contains the features of vertex $i$ (Liu & Zhou, 2020).

Crucially, graphs are defined up to relabeling of vertices. If $P$ is any $N \times N$ permutation matrix, the relabeled graph has adjacency $A' = PAP^\top$ and attributes $X' = PX$. These permutations leave the graph unchanged as an object; consequently, any model or inference procedure should be invariant to such relabeling, i.e., produce the same results for $(A, X)$ and $(A', X')$ (Joshi, 2025).

### 2.2 Amortized Bayesian Inference

Amortized Bayesian Inference (ABI) trains a single predictive model – implemented here as two jointly trained neural networks — that maps data $D$ to an approximate posterior over parameters $p(\theta \mid D)$. Instead of solving a new inference problem for each new dataset (e.g., via MCMC (Brooks et al., 2011) or variational inference (Blei et al., 2017)), ABI learns an inference network once and then reuses it for any new $D$ at negligible cost (Gershman & Goodman, 2014).

We generate simulated training pairs $(D, \theta)$ and learn (i) a summary network $h$ that compresses $D$ to a fixed-dimensional summary $s = h(D)$, and (ii) a conditional invertible neural network (cINN) $f_\phi$, also called inference network, that defines a bijection between parameters $\theta$ and a simple latent $z \sim \mathcal{N}(0, I)$ conditioned on $s$ (Radev et al., 2023b):

$$z \;=\; f_\phi(\theta; s), \qquad \theta \;=\; f_\phi^{-1}(z; s). \tag{1}$$

Our focus in this work is the design of the summary network, which must capture the specific characteristics of graph-structured data. A series of suitable deep network architectures will be presented in the following Section 2.3. For the inference network we use a standard generative architecture, such as coupling flows (Kingma & Dhariwal, 2018) or flow matching (Liu et al., 2022).

The inference network induces an approximate posterior $p_\phi(\theta \mid s)$ through the change of variables formula. For a conditional normalizing flow with coupling layers, the log-density takes the form

$$\log p_\phi(\theta \mid s) \; = \; \log p_Z\big(f_\phi(\theta; s)\big) \; + \; \log\Big|\det \tfrac{\partial f_\phi}{\partial \theta}(\theta; s)\Big|, \tag{2}$$

with $p_Z$ being the standard Gaussian density. Training minimizes the KL divergence between the true and approximate posteriors in expectation over the joint distribution $p(\theta, D) = p(\theta)\, p(D \mid \theta)$, which yields the Monte Carlo loss used in practice for a batch of $M$ simulated datasets and data-generating parameters $\{(D^{(m)}, \theta^{(m)})\}_{m=1}^M$:

$$\mathcal{L}(\phi, \psi) \; = \; \frac{1}{M} \sum_{m=1}^M -\log p_\phi\left(\theta^{(m)} \mid s = h(D^{(m)})\right). \tag{3}$$

After convergence, inference for an observed graph $D_{\text{obs}}$ proceeds by computing $s_{\text{obs}} = h(D_{\text{obs}})$ and then drawing posterior samples via $z^{(m)} \sim \mathcal{N}(0, I)$ and $\theta^{(m)} = f_\phi^{-1}\big(z^{(m)}; s_{\text{obs}}\big)$. To respect graph symmetries, $h$ is chosen permutation-invariant in the node indices so that $h(PAP^\top, PX) = h(A, X)$ for any permutation matrix $P$, ensuring the amortized posterior $q_\phi(\theta \mid h(x))$ is invariant to relabeling. In our experiments in Section 3 we instantiate the inference network $f_\phi$ as a coupling flow (Kingma & Dhariwal, 2018; Durkan et al., 2019; Ardizzone et al., 2021). We also tested flow matching variants (Liu et al., 2022; Lipman et al., 2023; Tong et al., 2024), which yielded comparable results. Different possibilities for the graph-aware summary architecture are introduced below.

### 2.3 Architectures suitable as summary networks

We require summary networks whose architectures respect the graph structure and are insensitive to the arbitrary ordering of nodes. Concretely, the model's intermediate layers should be permutation equivariant over node indices, and the final summary should be permutation invariant. In this paper we consider (i) message-passing Graph Neural Networks — specifically Graph Convolutional Networks (GCNs) (Kipf & Welling, 2017) — and (ii) transformer-based models suited to sets and graphs, namely the Set Transformer (Lee et al., 2019) and the Graph Transformer (Rampášek et al., 2023). These architectures aggregate information across neighborhoods or the whole graph while preserving the necessary permutation properties. As a simple baseline model we also compare them to the performance of Deep Sets (Zaheer et al., 2017). Table 1 provides an overview of the architectures considered in this work and points to the sections where each is described.

Table 1: Overview of the model architectures evaluated as summary networks in this work and the corresponding sections where they are introduced.

| Architecture | Section |
|---|---|
| Graph Convolutional Network | 2.3.1 |
| Deep Sets | 2.3.2 |
| Set Transformer | 2.3.4 |
| Graph Transformer | 2.3.5 |

### 2.3.1 Graph convolutional network

**Neural message passing.** A graph neural network (GNN) updates a hidden embedding $\mathbf{h}_i^{(k)}$ for each node $i \in V$ by aggregating information from its graph neighborhood $\mathcal{N}(i) \subseteq V$. In iteration (layer) $k$, a message $\mathbf{m}_{\mathcal{N}(i)}^{(k)}$ for node $i$ is formed from the embeddings of neighboring nodes $j \in \mathcal{N}(i)$. The new embedding $\mathbf{h}_i^{(k+1)}$ is obtained by combining the previous embedding $\mathbf{h}_i^{(k)}$ with the message $\mathbf{m}_{\mathcal{N}(i)}^{(k)}$. As initial embeddings one can choose the node attributes $\mathbf{x}_i$; if no attributes are available, simple structural statistics (e.g., degree or centrality) can be used. After $k$ iterations, node $i$ has incorporated information from its $k$-hop neighborhood (Hamilton, 2020).

**The basic GNN.** A simplified version of GNN models (Merkwirth & Lengauer, 2005; Scarselli et al., 2009) is

$$\mathbf{h}_i^{(k)} = \sigma\left( \mathbf{W}_{\text{self}}^{(k)} \mathbf{h}_i^{(k-1)} + \mathbf{W}_{\text{neigh}}^{(k)} \sum_{j \in \mathcal{N}(i)} \mathbf{h}_j^{(k-1)} + \mathbf{b}^{(k)} \right), \tag{4}$$

with trainable parameter matrices $\mathbf{W}_{\text{self}}^{(k)}, \mathbf{W}_{\text{neigh}}^{(k)} \in \mathbb{R}^{d^{(k)} \times d^{(k-1)}}$, bias $\mathbf{b}^{(k)} \in \mathbb{R}^{d^{(k)}}$, and a non linear activation function $\sigma$.

**Graph Convolutional Network.** Compared to the basic GNN, the Graph Convolutional Network (GCN) adds self-loops (treating the node itself as a neighbor) and applies symmetric degree normalization to reduce sensitivity to node degree and improve stability. Writing $\tilde{\mathcal{N}}(i) = \mathcal{N}(i) \cup \{i\}$ for the self-loop–augmented neighborhood, the update becomes (Kipf & Welling, 2017; Hamilton, 2020):

$$\mathbf{h}_i^{(k)} = \sigma\left( \mathbf{W}^{(k)} \sum_{j \in \tilde{\mathcal{N}}(i)} \frac{\mathbf{h}_j^{(k-1)}}{\sqrt{|\tilde{\mathcal{N}}(i)| \, |\tilde{\mathcal{N}}(j)|}} + \mathbf{b}^{(k)} \right). \tag{5}$$

### 2.3.2 Deep Sets

As a standard neural architecture, we adopt Deep Sets as a baseline (Zaheer et al., 2017). For a graph $G = (V, E)$ with node attributes $\{\mathbf{x}_i\}_{i \in V}$, Deep Sets define a permutation-invariant function on the set of node features,

$$f(\{\mathbf{x}_i : i \in V\}) = \rho\left( \sum_{i \in V} \phi(\mathbf{x}_i) \right), \tag{6}$$

where the element encoder $\phi$ and the set decoder $\rho$ are standard neural networks shared across elements. The sum aggregation makes $f$ invariant to any reordering of nodes, which is appropriate since node indices have no intrinsic order.

In contrast to graph-specific neural networks such as the Graph Convolutional Network, a plain Deep Sets model does not use the adjacency $\mathbf{A}$ and therefore cannot exploit the explicit graph topology (e.g., neighborhoods, paths). It effectively acts as a "bag-of-nodes" baseline that captures the multiset of node attributes but ignores the edges. This simplicity yields an efficient and robust point of comparison in our setting: implementation reduces to two small MLPs and a permutation-invariant pooling.

In the experiments in Section 3 we often consider graphs with a fixed number of nodes $N$. In this case, the adjacency matrix $\mathbf{A} \in \{0, 1\}^{N \times N}$ (or weighted) can be incorporated into a Deep Sets baseline by augmenting each node's features with its adjacency row. Concretely,

$$D_i = \left[ \mathbf{x}_i \,;\, \mathbf{A}_{i,:} \right]. \tag{7}$$

The Deep Sets then operates on the set $\{D_i\}_{i \in V}$

$$f(\{D_i : i \in V\}) = \rho\left( \sum_{i=1}^{N} \phi(D_i) \right). \tag{8}$$

This preserves permutation invariance under any simultaneous relabeling of nodes (i.e., $\mathbf{A}' = \mathbf{P}\mathbf{A}\mathbf{P}^\top$, $\mathbf{x}_i' = \mathbf{x}_{\pi(i)}$): rows and features permute together, and the sum aggregation is order-agnostic.

### 2.3.3 Attention, transformers, and their application to graphs

We also employ transformer-based models for graph-structured data. Below we summarize scaled dot-product attention, multi-head attention, the transformer layer, and why these mechanisms are natural for graphs.

**Scaled dot-product attention.** Let $\mathbf{X} = [\mathbf{x}_1; \dots; \mathbf{x}_n] \in \mathbb{R}^{n \times d_{\text{model}}}$ denote $n$ input vectors (tokens or nodes) with $d_{\text{model}}$ as the input dimension. Attention is a content-based weighting mechanism that lets each element $\mathbf{x}_i$ aggregate information from all elements (Vaswani et al., 2017). Let $\mathbf{Y} \in \mathbb{R}^{n \times d_{\text{model}}}$ denote the set of vectors being attended to. The attention operation is parameterized through queries, keys, and values,

$$\mathbf{Q} = \mathbf{X}\mathbf{W}^Q, \quad \mathbf{K} = \mathbf{Y}\mathbf{W}^K, \quad \mathbf{V} = \mathbf{Y}\mathbf{W}^V, \tag{9}$$

with projection matrices $\mathbf{W}^Q, \mathbf{W}^K \in \mathbb{R}^{d_{\text{model}} \times d_k}$ and $\mathbf{W}^V \in \mathbb{R}^{d_{\text{model}} \times d_v}$. When $\mathbf{X} = \mathbf{Y}$, the mechanism reduces to self-attention; otherwise, it is referred to as cross-attention. Writing $\mathbf{q}_i$, $\mathbf{k}_j$, $\mathbf{v}_j$ for the $i$th/$j$th rows of $\mathbf{Q}$, $\mathbf{K}$, $\mathbf{V}$, the attention weights from $j$ to $i$ are

$$\alpha_{ij} = \text{softmax}_j \left( \frac{\mathbf{q}_i \mathbf{k}_j^\top}{\sqrt{d_k}} \right), \tag{10}$$

which satisfy $\alpha_{ij} \geq 0$ and $\sum_j \alpha_{ij} = 1$. The output for element $i$ is the convex combination $\mathbf{z}_i = \sum_{j=1}^n \alpha_{ij} \mathbf{v}_j$. In matrix form this results in

$$\text{Attn}(\mathbf{Q}, \mathbf{K}, \mathbf{V}) = \text{softmax}\left( \frac{\mathbf{Q}\mathbf{K}^\top}{\sqrt{d_k}} \right) \mathbf{V}, \tag{11}$$

where the softmax is applied row-wise. The factor $1/\sqrt{d_k}$ stabilizes training by controlling score magnitudes (Vaswani et al., 2017). This realizes differentiable, data-dependent routing: the model learns *what* to attend to (via $\mathbf{W}^Q, \mathbf{W}^K$) and *how much* to pass (via $\alpha_{ij}$), enabling global interactions without fixed receptive fields.

**Multi-head attention (MHA).** MHA repeats attention in $H$ learned subspaces (heads) (Vaswani et al., 2017). For head $\ell \in \{1, \dots, H\}$,

$$\mathbf{Q}_\ell = \mathbf{X}\mathbf{W}_\ell^Q, \quad \mathbf{K}_\ell = \mathbf{Y}\mathbf{W}_\ell^K, \quad \mathbf{V}_\ell = \mathbf{Y}\mathbf{W}_\ell^V, \qquad \text{head}_\ell = \text{Attn}(\mathbf{Q}_\ell, \mathbf{K}_\ell, \mathbf{V}_\ell), \tag{12}$$

and the heads are concatenated and projected to

$$\text{MHA}(\mathbf{X}, \mathbf{Y}, \mathbf{Y}) = \text{Concat}(\text{head}_1, \dots, \text{head}_H) \mathbf{W}^O, \qquad \mathbf{W}^O \in \mathbb{R}^{(H d_v) \times d_{\text{model}}}. \tag{13}$$

A common choice is $d_k = d_v = d_{\text{model}}/H$. Distributing capacity across heads helps capture heterogeneous relations at different granularities; heads often specialize, though redundancy is also observed (Clark et al., 2019; Voita et al., 2019; Michel et al., 2019; Shazeer et al., 2020).

**Transformer layers.** Transformers stack MHA with position-wise feed-forward networks (FFN), each sublayer wrapped with residual connections and layer normalization (LN) (Vaswani et al., 2017). An encoder layer is defined by

$$\mathbf{H}' = \text{MHA}(\text{LN}(\mathbf{H})) + \mathbf{H}, \qquad \mathbf{H}_{\text{out}} = \text{FFN}(\text{LN}(\mathbf{H}')) + \mathbf{H}', \tag{14}$$

and applied in parallel over positions. Decoder layers additionally contain (i) masked self-attention for autoregressive generation and (ii) cross-attention conditioning on encoder outputs.

**Why attention for graphs?** For a graph with nodes $V = \{1, \dots, n\}$ and features $\mathbf{x}_i$, self-attention can be viewed as message passing on a fully connected graph: node $i$ aggregates messages $\mathbf{v}_j$ from all nodes $j$ with adaptive, data-dependent edge weights $\alpha_{ij}$ learned from $(\mathbf{q}_i, \mathbf{k}_j)$ (Joshi, 2025). Thus each layer has a global receptive field. At the same time, one can incorporate standard graph inductive biases without

explicitly hard-coding the graph topology. This is commonly done in two ways. First, structural or positional information, such as shortest-path distances, random-walk features, or Laplacian eigenvectors, is added to the node representations or the attention queries/keys. Second, attention masks or bias terms are introduced to discourage or encourage attention between certain node pairs. These mechanisms can recover GNN-like locality when desired, while still leveraging the Transformer's capacity to model long-range, multi-hop dependencies with a single layer (Dwivedi & Bresson, 2021),

### 2.3.4 Set Transformer

The Set Transformer (Lee et al., 2019) utilizes the attention mechanism above while remaining permutation-invariant like Deep Sets. It comprises an encoder followed by a decoder; each layer applies attention and uses a parameterized pooling mechanism.

To formalize self-attention on sets, define the Multi-head Attention Block (MAB) (Lee et al., 2019) for $\mathbf{X}, \mathbf{Y} \in \mathbb{R}^{n \times d}$:

$$\mathrm{MAB}(\mathbf{X}, \mathbf{Y}) = \mathrm{LN}\big(\mathbf{H} + \mathrm{rFF}(\mathbf{H})\big), \quad \text{with} \quad \mathbf{H} = \mathrm{LN}\big(\mathbf{X} + \mathrm{MHA}(\mathbf{X}, \mathbf{Y}, \mathbf{Y})\big). \tag{15}$$

Here, $\mathrm{MHA}(\mathbf{Q}, \mathbf{K}, \mathbf{V})$ denotes multi-head attention with arbitrary queries/keys/values as defined in 13. The function rFF is a row-wise feed-forward network applied independently to each row.

The Self-Attention Block (SAB) is then defined as

$$\mathrm{SAB}(\mathbf{X}) = \mathrm{MAB}(\mathbf{X}, \mathbf{X}). \tag{16}$$

To reduce the $\mathcal{O}(n^2)$ cost of full self-attention, the Induced Set Attention Block (ISAB) (Lee et al., 2019) introduces $r$ learned inducing points $\mathbf{I} \in \mathbb{R}^{r \times d}$ with $r \ll n$:

$$\mathbf{H} = \mathrm{MAB}(\mathbf{I}, \mathbf{X}), \qquad \mathrm{ISAB}(\mathbf{X}) = \mathrm{MAB}(\mathbf{X}, \mathbf{H}), \tag{17}$$

which yields $\mathcal{O}(nr)$ complexity.

For pooling, instead of a fixed aggregator, Pooling by Multi-head Attention (PMA) is used. Let $\mathbf{Z} \in \mathbb{R}^{n \times d}$ be the encoder's output and $\mathbf{S} \in \mathbb{R}^{k \times d}$ a learned set of $k$ seed vectors:

$$\mathrm{PMA}_k(\mathbf{Z}) = \mathrm{MAB}(\mathbf{S}, \mathrm{rFF}(\mathbf{Z})) \in \mathbb{R}^{k \times d}. \tag{18}$$

Choosing $k = 1$ yields a single permutation-invariant set embedding; $k > 1$ produces multiple summaries (e.g., for clustering).

A typical Set Transformer instantiation is

$$\mathbf{Z} = \mathrm{Encoder}(\mathbf{X}) = \mathrm{SAB}\big(\mathrm{SAB}(\mathbf{X})\big), \tag{19}$$

$$\mathrm{Decoder}(\mathbf{Z}) = \mathrm{rFF}\big(\mathrm{SAB}\big(\mathrm{PMA}_k(\mathbf{Z})\big)\big) \in \mathbb{R}^{k \times d}. \tag{20}$$

This architecture remains permutation-invariant (for $k = 1$) or permutation-equivariant (for per-element outputs) by construction while allowing rich interactions among set elements via attention (Lee et al., 2019).

### 2.3.5 Graph Transformer

A Graph Transformer keeps the standard Transformer building blocks — multi-head attention and position-wise feed-forward networks (rFF) — but makes attention graph aware (Vaswani et al., 2017; Dwivedi & Bresson, 2021). The core idea is simple: compute the usual scaled dot-product weights $\alpha_{ij} = \mathrm{softmax}_j\big(\mathbf{q}_i \mathbf{k}_j^\top / \sqrt{d_k}\big)$, but restrict these scores using the graph structure. In our case, we use an adjacency mask with self-loops, so each node attends only to its neighbors (and itself) in a given layer (Veličković et al., 2018). Each layer follows a pre-norm encoder design with residual connections for stability: $\mathrm{LN} \to \mathrm{MHA} \to$ residual and $\mathrm{LN} \to \mathrm{FFN} \to$ residual (Vaswani et al., 2017; Xiong et al., 2020). For graph-level outputs, we use Pooling by Multi-head Attention (PMA) from the Set Transformer (Eq. (18)) (Lee et al., 2019). Stacking several masked-attention layers yields multi-hop information flow, while PMA provides a learnable alternative to fixed sum/mean pooling. This contrasts with global graph Transformers that attend over all node pairs and inject topology via positional/structural encodings or attention biases rather than strict masks (Dwivedi & Bresson, 2021; Rampášek et al., 2023).

## 2.4 Evaluation metrics

In the experiments (Section 3), we instantiate the ABI pipeline with each of the proposed architectures as the summary network across a range of problem settings. To enable a fair comparison and to quantify posterior quality, we report three complementary metrics — calibration, posterior contraction, and recovery — assessing, respectively, coverage reliability, concentration of uncertainty, and correspondence between true parameters and posterior summaries (Lueckmann et al., 2021; Hermans et al., 2022). In all experiments, metrics are computed on unseen simulated test data.

### 2.4.1 Simulation-based Calibration

Simulation-Based Calibration (SBC, Cook et al. (2006); Talts et al. (2020); Modrák et al. (2025)) assesses whether a Bayesian inference pipeline (model, simulator, and inference algorithm) yields calibrated posteriors for the examined parameters separately — i.e., credible intervals with correct long-run frequency. The key idea is that if we sample parameters from the prior, generate data from the model, and then infer the posterior, the true sampled parameter should look like a draw from that posterior.

To check this visually, we repeat it for multiple parameter draws from the prior and summarize each repetition by the posterior CDF value of the true parameter. If posteriors are well calibrated, these values should be uniformly spread across the unit interval. The ECDF is plotted against a uniform ECDF and drawn with simultaneous bands (Säilynoja et al., 2022). Systematic departures are diagnostic: concave curves indicate under-dispersion (overconfident posteriors), convex curves indicate over-dispersion, and vertical shifts or tilts signal bias.

To complement the ECDF, we report a single-number summary that tests for any departure from uniformity across all rank cutpoints. Following Modrák et al. (2025), define

$$\gamma = 2 \min_{i \in \{1,...,M+1\}} \left( \min\{\text{Bin}(R_i \mid S, z_i), 1 - \text{Bin}(R_i - 1 \mid S, z_i)\} \right) , \tag{21}$$

where $M$ is the number of posterior draws used to compute ranks per replication, $S$ is the number of SBC replications, $z_i = i/(M+1)$ is the expected fraction of ranks strictly less than $i$ under perfect calibration, $R_i$ is the observed count of such ranks across $S$ replications, and $\text{Bin}(R \mid S, p)$ is the binomial CDF with $S$ trials and success probability $p$ evaluated at $R$ (Säilynoja et al., 2022).

Following the suggestion from Säilynoja et al. (2022), we calibrate it against its null distribution (obtained by simulating uniform ranks given $M$ and $S$) and let $\bar{\gamma}$ denote the 5th percentile of that null distribution to make this statistic comparable across settings. We then report the score

$$\ell_\gamma := \log \left( \frac{\gamma}{\bar{\gamma}} \right) . \tag{22}$$

Values below zero imply that the observed deviation is more extreme than 95% of null realizations, i.e., rejection of uniformity at the 5% level and thus evidence of miscalibration. This scalar complements the ECDF by summarizing global calibration in a single, interpretable statistic (Säilynoja et al., 2022).

### 2.4.2 Posterior Contraction

Posterior contraction quantifies the reduction in uncertainty achieved when updating the prior to the posterior (Schad et al., 2020). We define

$$\text{PC} := 1 - \frac{\sigma^2_{\text{post}}}{\sigma^2_{\text{prior}}} , \tag{23}$$

where $\sigma^2_{\text{post}}$ and $\sigma^2_{\text{prior}}$ denote the posterior and prior variances, respectively. Values of PC close to 1 indicate strong contraction (substantial uncertainty reduction), whereas values near 0 indicate little or no change in variance for the marginal parameter under consideration.

### 2.4.3 Recovery

To quantify parameter recovery (PR), we compute the Pearson correlation coefficient (Rodgers & Nicewander, 1988) between the ground-truth parameter values and the medians of the corresponding posterior samples. Note that any reduction of a posterior to a single point estimate, here the median, can be misleading, especially when the posterior is multi-modal.

In addition, we report diagnostics plots in which the true parameter values are shown against the posterior medians, with vertical bars indicating 95% credible intervals. These plots make it easier to detect systematic bias and distributional asymmetry and skewness that are not visible from the point estimate alone.

## 3 Experiments

We conduct three experiments to compare neural architectures used as summary networks. The first experiment is a controlled toy problem, in which we evaluate 12 summary-network variants on amortized posterior estimation. The second experiment is a biologically motivated case study that models the interaction network of free-ranging mice. The third experiment considers a logistics application and evaluates model performance in a neural likelihood estimation setting. Since none of the three experiments have a tractable likelihood, direct comparison with non-SBI baselines such as MCMC is not possible. To give an indication of how our ABI approach compares to such a baseline, we present a simplified example with a tractable likelihood in Appendix C. All code and material for the experiments can be found on GitHub[1].

### 3.1 Toy example: Estimating probabilities of node connections

In the first experiment, we study a controlled setting with only four graph-level parameters of interest. This design is well suited for an initial comparison of summary-network architectures: one parameter is intentionally more complex, providing a challenging test case that highlights differences in performance across models.

### 3.1.1 Problem setup

We begin with a controlled toy example in which we simulate an undirected graph with $N = 30$ nodes. In Section 3.1.4, we increase the difficulty of the inference task by varying the graph size; here, however, we focus on the fixed-size setting. Each node is labeled either $A$ or $B$. The number of $A$-nodes, `num_a`, is drawn from a discrete uniform distribution `num_a` $\sim \text{Unif}\{5, 6, \ldots, 25\}$, and the remaining $N - $ `num_a` nodes are labeled $B$. Let $X \in \{0, 1\}^N$ denote the node-feature vector (e.g., $X_i = 1$ if node $i$ is type $A$, else 0).

Conditional on the labels, edges are sampled independently (no self-loops) with type-specific probabilities $\pi_{AA}, \pi_{BB}, \pi_{AB} \sim \text{Unif}(0.1, 0.9)$. Node $i$ and node $j$ are connected via an edge with a probability of

$$p_{ij} = \begin{cases} \pi_{AA}, & X_i = X_j = 1, \\ \pi_{BB}, & X_i = X_j = 0, \\ \pi_{AB}, & X_i \neq X_j. \end{cases} \tag{24}$$

This yields a symmetric adjacency matrix $A \in \{0, 1\}^{N \times N}$ with a zero diagonal.

To increase structural complexity, we introduce a triadic-closure parameter $\lambda \sim \text{Unif}(0.1, 0.9)$, which alters the probability of an edge between two nodes when they share common neighbors. Let $\text{CN}_{ij}$ be the number of common neighbors of nodes $i$ and $j$, computed from $A$. For each pair $(i, j)$ with $A_{ij} = 0$, we revisit the potential edge whenever $\text{CN}_{ij} > 0$. In that case, we sample the edge according to $\tilde{A}_{ij} \sim \text{Bernoulli}(\lambda)$, and set $\tilde{A}_{ji} = \tilde{A}_{ij}$, leaving all other entries unchanged. The resulting matrix $\tilde{A}$ constitutes the final output of the simulator. The parameters of interest are $\theta = (\pi_{AA}, \pi_{BB}, \pi_{AB}, \lambda)$, for which we perform neural posterior inference.

---

[1]https://github.com/sjedhoff/ABI-graph-paper

### 3.1.2 Training architecture

We use the proposed ABI framework to enable posterior inference on the parameters $\theta$. The summary network takes the simulated graph with the adjacency matrix and the feature vector $(\tilde{A}, X)$ as input and produces a fixed-length summary vector. In this case study, we use a 16-dimensional summary, which exceeds the parameter dimension of four and follows the heuristic that the summary dimension should not be smaller than the number of target parameters (Li et al., 2025). We evaluate four summary-network architectures: a Graph Convolutional Network (GCN), Deep Sets, a Graph Transformer, and a Set Transformer. For each architecture, we test three pooling mechanisms: simple mean pooling as a baseline, the permutation-invariant layer from Deep Sets (see Eq. 8), and, as the most expressive option, pooling via multi-head attention (PMA) (see Eq. 18). The inference network takes the summary $s = h(D)$ as input and outputs an approximate posterior $p_\phi(\theta \mid s)$ over $\theta$. We use a coupling flow with spline transformations built with 6 invertible layers. Each model is trained online for 250 epochs, with 100 batches per epoch and a batch size of 32.

### 3.1.3 Results

Figure 2 summarizes the experimental results. For better visualization, the three parameters $\pi_{AA}, \pi_{BB}$, and $\pi_{AB}$ are presented together in the first row labeled $\pi$. Table 2 in Appendix A provides additional details on the results, including the final training loss and the number of trainable parameters for each summary network architecture. Table 3 in the same appendix reports training times across different dataset sizes, and Figure 11 illustrates how dataset size affects model performance.

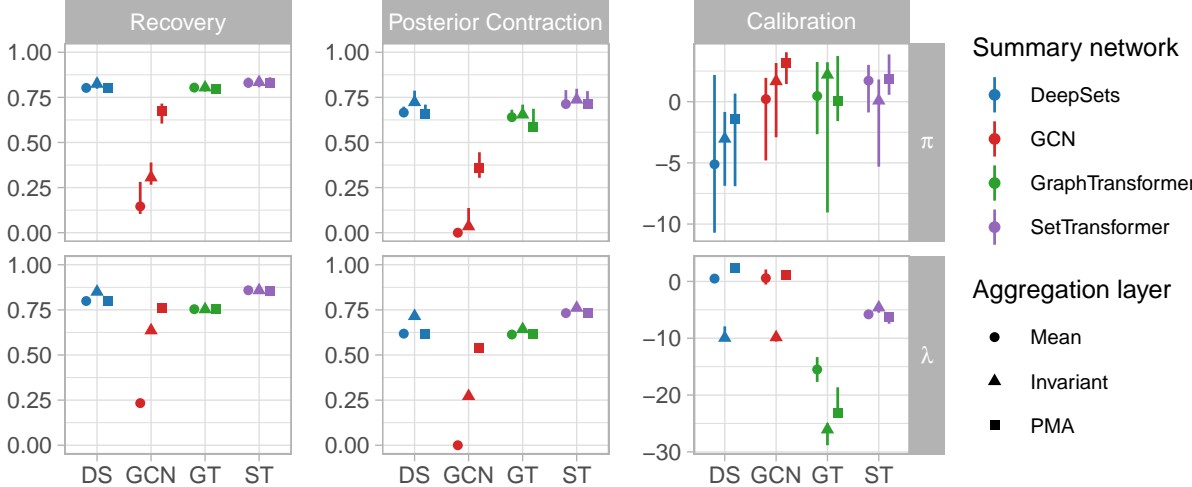

Figure 2: Results for the toy example across four summary-network architectures with each three different aggregation layers. We report parameter recovery (higher is better), posterior contraction (higher is better), and calibration ($\ell_\gamma$; values $> 0$ indicate good calibration) for the parameters $\pi = \{\pi_{AA}, \pi_{BB}, \pi_{AB}\}$ and $\lambda$. Markers (circles/triangles/squares) indicate the median across five runs, and error bars span the minimum to maximum values.

The $\pi$ parameters, which represent the base edge probabilities between $A$ and $B$ nodes, are easy to recover: almost all models achieve recovery scores above 0.8. The triadic-closure parameter $\lambda$ shows broadly comparable recovery across the Deep Sets, Graph Transformer and Set Transformer architectures. In contrast, the Graph Convolutional Network — particularly with the simpler aggregation layers — fails to achieve satisfactory parameter recovery for both $\pi = \{\pi_{AA}, \pi_{BB}, \pi_{AB}\}$ and $\lambda$. The other three architectures recover the parameters adequately across all aggregation layers. Posterior contraction follows a similar pattern, with the Graph Convolutional Network performing worse than Deep Sets, the Graph Transformer, and the Set Transformer; the Set Transformer achieves the strongest posterior contraction throughout.

Achieving adequate calibration is substantially more challenging, as shown in the right column of Figure 2. For the $\pi$ parameters, the Graph Convolutional Network is well calibrated, whereas the Deep Sets fails to achieve adequate calibration. This can be explained by the high recovery attained by Deep Sets, which makes good calibration harder to obtain. For $\lambda$, the Graph Transformer shows the weakest recovery, while Deep Sets and the Graph Convolutional Network, each with mean and PMA aggregation, are well calibrated. The Set Transformer does not achieve adequate calibration, with $\ell_\gamma$ values around $-5$. This may again be a consequence of its high parameter recovery (approximately 0.86), which makes achieving good calibration more difficult.

Figure 3 compares a representative run of the Graph Convolutional Network with mean pooling and the Set Transformer with multi-head attention pooling in terms of recovery and calibration. The recovery plot (left) shows estimates for $\pi_{AB}$ as a representative example of the three $\pi$ parameters. For both $\pi_{AB}$ and $\lambda$, the Graph Convolutional Network exhibits poorer recovery than the Set Transformer. Vertical lines denote 95% credible intervals, which are noticeably narrower for the Set Transformer, indicating sharper estimates. The Graph Convolutional Network appears to learn mainly the prior distribution, a common failure mode, whereas the Set Transformer learns a meaningful posterior, which can be seen by a correlation of 0.873 and 0.847 for $\pi_{AB}$ and $\lambda$ respectively of posterior-median and true parameter value. The ECDF calibration plots (right panel of Figure 3) provide a diagnostic of posterior calibration based on simulation-based ranks, as described in Section 2.4. Both the Graph Convolutional Network and the Set Transformer calibrate the parameter adequately, as their rank ECDFs lie within the 95% confidence bands. Model performance can be assessed not only through parameter-based metrics, but also through data-dependent quantities. Appendix B presents data-dependent SBC results for this experiment.

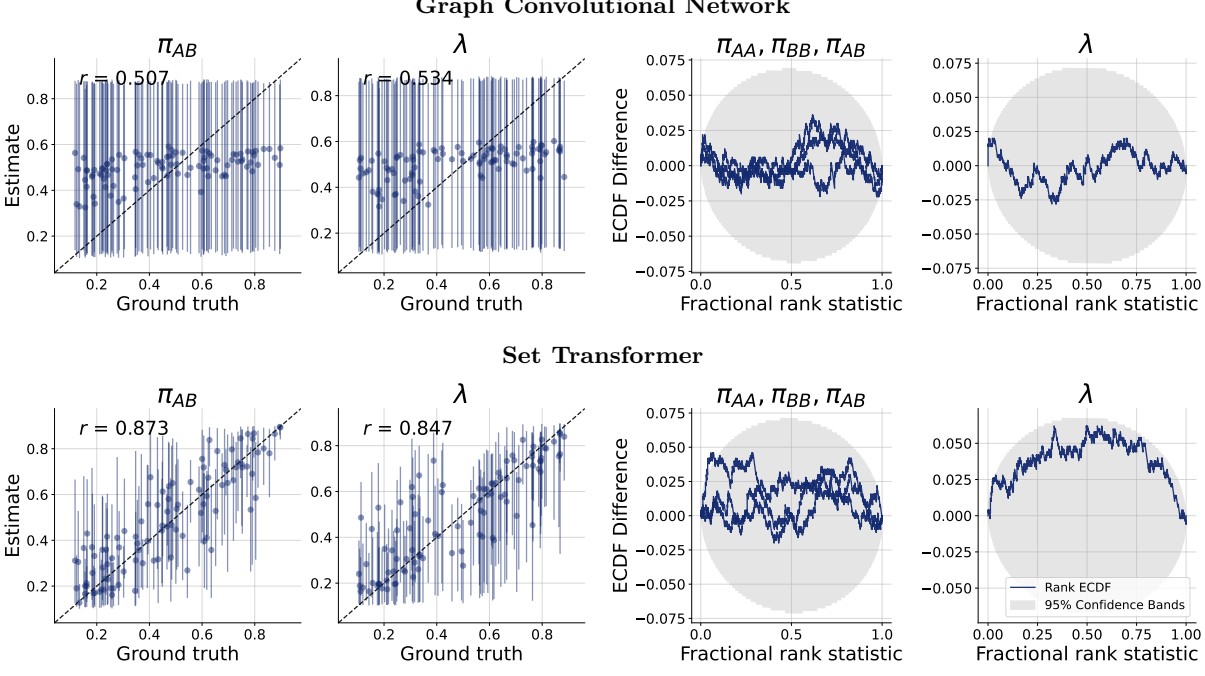

Figure 3: Recovery (median and 95% credible interval) and Calibration (ECDF Difference plots) of parameters for one run of the Graph Convolutional Network (with mean pooling) and Set Transformer (with multi-head attention pooling).

This case study indicates that the Set Transformer performs best overall, achieving the strongest parameter recovery, posterior contraction, and calibration in most settings. The Deep Sets baseline is also competitive, often yielding comparable results. In contrast, the two explicitly graph-structured architectures — the Graph Convolutional Network and the Graph Transformer — do not outperform Deep Sets, despite leveraging the adjacency information.

This suggests that the chosen toy example may not require rich structural reasoning: much of the relevant information may be captured by node features and global aggregation alone. Consequently, incorporating $k$-hop message passing via a Graph Convolutional Network may not provide a clear advantage in this setting.

### 3.1.4 Results for varying sizes of graph

To demonstrate that the approach also handles graphs of varying size, we extend the toy setup by drawing the number of nodes uniformly from $N \in \{10, \dots, 50\}$, while keeping all other components unchanged. To accommodate variable-sized inputs within our fixed-size implementation, we pad both the adjacency matrix $A$ and the node-feature vector $X$. Because the Set Transformer with PMA performed best in the previous experiment, we focus on this configuration in the variable-$N$ setting. We again train the Set Transformer summary network jointly with a coupling-flow inference network using spline transformations for 250 epochs of online training.

Figure 4 shows recovery results for $\pi_{AB}$ and $\lambda$, comparing graphs with $N = 15$ nodes (left) and $N = 45$ nodes (right). For both graph sizes, $\pi_{AB}$ is recovered accurately. However, the 95% credible intervals (vertical error bars) are slightly narrower for $N = 45$; the median interval width for $N = 15$ is about 7% larger than for $N = 45$, indicating reduced posterior uncertainty. For $\lambda$, the credible intervals also shrink by roughly 30% and parameter recovery, measured as the correlation between posterior medians and ground-truth values, improves for the larger graphs.

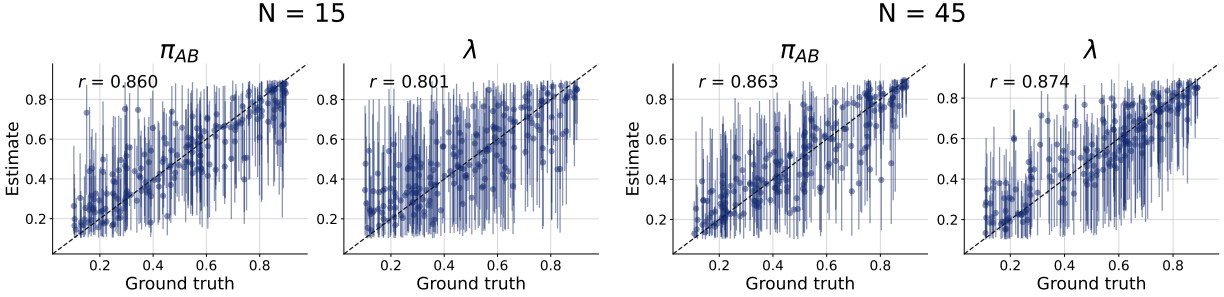

Figure 4: Recovery parameters $\pi_{AB}$ and $\lambda$, shown as posterior median and 95% credible intervals, for graphs with $N = 15$ nodes (left) and $N = 45$ nodes (right). Posterior samples are drawn from the same model (Set Transformer with PMA as the summary network), trained on graphs with $N \in \{10, \dots, 50\}$.

Overall, performance is stronger for larger graphs, which is expected: with more nodes (and thus more potential edges and triadic structures), the data contain more information about the underlying parameters, leading to more precise inference. Moreover, the combination of a Set Transformer with PMA as the summary network and a spline-based coupling flow as the inference network successfully amortizes inference across varying graph sizes, with no observable performance degradation relative to the fixed-size setting.

### 3.2 Mice interaction network

In this case study, we consider a biological application: how social interactions among free-ranging mice shape their gut microbiome. We focus on the composition and dynamics of the gut microbiome under microbial transmission through social contact (Raulo et al., 2024). The experimental setup and simulator used in the following are grounded in a real-world system with observed data (Raulo et al., 2023). Specifically, the study concerns wild wood mice (Apodemus sylvaticus), a semi-social rodent that forms structured social networks comprising both strong and weak ties; many individuals share no direct ties, yielding a sparse network. Social networks are constructed from spatiotemporal co-occurrence frequencies estimated from tracking data. In the original study, roughly 150 mice in a $\sim 2.5$ ha area were implanted with subcutaneous RFID (PIT) tags and released at their capture locations. These tags enabled continuous tracking of individual movements, which was then used to quantify spatial overlap and social associations between mouse pairs.

In the original data collecting process, in addition to behavioral data, each mouse's gut microbiome was characterized from fecal samples collected upon trapping. DNA was extracted from the feces, the bacterial 16S rRNA marker gene was sequenced, and the resulting sequence variants were taxonomically assigned using the Silva 138 bacterial reference database (see Raulo et al. (2024) for methodological details and Raulo et al. (2023) for the data).

Our goal is to use a simulator to study how the composition of microbial taxa in each mouse's gut changes over time, and how similarity in community composition relates to the pattern and strength of social associations. Because direct contact and close spatial co-occurrence both facilitate microbial transfer, we expect mice that interact frequently to become increasingly similar in their microbiome composition over sufficiently long time scales.

### 3.2.1 Simulator setup

To investigate these dynamics, we developed a simulator that represents a cohort of mice as a weighted interaction network $G = (V, E)$. Each mouse is represented by a node $i \in V$, and edges $e_{ij} \in E$ capture interaction propensity based on spatial proximity, temporal co-occurrence, and/or social association. Edge weights $w_{ij} \in [0, 1]$ quantify expected daily interaction intensity, while a global network-density parameter $\delta$ controls the overall number of ties in the graph. At initialization, every mouse $i$ is assigned a random subset of microbial taxa, stored as node attributes. The simulator then proceeds in discrete daily steps. On each day, mouse pairs $(i, j)$ interact whenever $w_{ij} > 0$; conditional on the interaction, they exchange microbiota in proportion to both the edge weight $w_{ij}$ and an exchange factor $\alpha \in [0, 1]$, which governs the amount of taxa transferred between partners. Repeating this process over multiple days leads to gradual convergence of community compositions among strongly connected mice, whereas weakly connected or isolated individuals retain more distinct microbiomes. Depending on network density and the exchange rate, the system approaches a steady state after several days.

In this case study, we simulate a group of 30 mice. Each mouse is initialized with 5 taxa drawn from a pool of 20 possible taxa. Because taxa are mouse-specific, we encode their abundances in the node-feature matrix $X \in \mathbb{R}^{30 \times 20}$, where each row corresponds to a mouse and each column to a taxon. Absent taxa are represented by zero; present taxa are stored as relative amounts that sum to 100% per mouse. Taxa whose amount falls below a presence threshold, here 0.0001%, are removed from a mouse's microbiome. A taxon is also removed if it is not exchanged for four days in a row.

Our goal is to infer two parameters — the network density $\delta$ and the exchange factor $\alpha$ — from (i) the adjacency matrix of the interaction network and (ii) the final-day microbial composition of each mouse, given by $X$. As priors we choose $\delta \sim \text{Unif}(0.01, 0.5)$ and $\alpha \sim \text{Unif}(0.05, 0.5)$. We run the simulator for 5, 10 and 30 days to assess how parameter recovery depends on the length of the observation horizon.

### 3.2.2 Training architecture

As summary networks, we evaluate the four architectures introduced in Section 2.3: Deep Sets (with invariant pooling), a Graph Convolutional Network (with invariant pooling), a Graph Transformer, and a Set Transformer (both using multi-head attention pooling). We fix the summary dimension to 16. The inference network is a coupling flow with spline-based transformations built with 6 layers. For each observation horizon (5, 10, and 30 days), we generate 50 000 simulations for training and an additional 1000 simulations for validation. All models use dropout with rate 0.05 and apply early stopping based on the validation loss to mitigate overfitting.

We compare summary networks using three metrics: the $\ell_\gamma$ score for calibration, posterior contraction (PC), and parameter recovery (PR), each reported separately for the network density ($\delta$) and the exchange factor ($\alpha$). Every experiment is repeated five times, and we report the median across runs.

### 3.2.3 Results

The results across summary-network architectures and the three observation horizons are summarized in Figure 5. Table 4 in Appendix D shows the exact median values of the metrics together with the final validation loss. The network-density parameter $\delta$ is recovered reliably: in most settings, recovery, measured as the correlation between posterior medians and ground-truth values, exceeds 0.9. Posterior contraction likewise indicates that the inferred posteriors for $\delta$ are suitably concentrated. However, the calibration score $\ell_\gamma$ reveals that, on average, none of the models achieves well-calibrated posteriors for $\delta$.

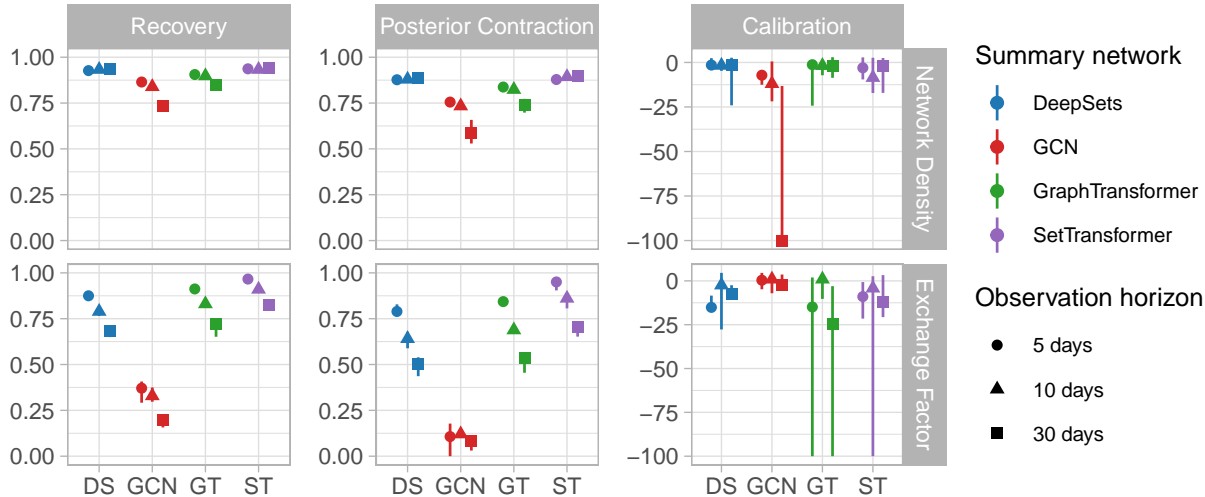

Figure 5: Results for the mice interaction network case study across four summary-network architectures, each evaluated for three different observation horizons. We report parameter recovery (higher is better), posterior contraction (higher is better), and calibration ($\ell_\gamma$; values $> 0$ indicate good calibration) for the parameters network density $\delta$ and exchange factor $\alpha$. Markers (circles/triangles/squares) indicate the median across five runs, and error bars span the minimum to maximum values.

Across all horizons, the Set Transformer performs best overall, with the strongest parameter recovery and posterior contraction for both $\delta$ and $\alpha$. For the exchange factor $\alpha$, Deep Sets, the Graph Transformer, and the Set Transformer achieve acceptable recovery, although recovery decreases as the observation horizon increases. The Graph Convolutional Network performs substantially worse, never exceeding a recovery of 0.37; posterior contraction for $\alpha$ mirrors this pattern.

Figure 6 illustrates recovery (top row) and calibration (bottom row) for 5-day (left) and 30-day (right) horizons from representative runs using the Set Transformer. For $\delta$, recovery is similar across horizons: it degrades at very low and very high true densities, but remains strongest at intermediate values (approximately 0.2–0.3). For $\alpha$, recovery is weaker at larger true values and improves for smaller ones, an effect that is more pronounced at 30 days. The ranked ECDF plots indicate good calibration for both parameters at Day 5, whereas at Day 30 the posterior for $\alpha$ shows a tendency toward overestimation.

The dependence of posterior quality and sharpness (reflected in the width of the 95% credible intervals) on the true parameter values is intuitive for the exchange factor $\alpha$. Large exchange factors drive the system to a steady state quickly, and once compositions have converged, additional daily exchanges produce little measurable change. With longer observation windows (e.g., 30 days), this steady-state regime dominates the data, making it harder to infer the true magnitude of exchange with only given measurements of the taxa of one single day.

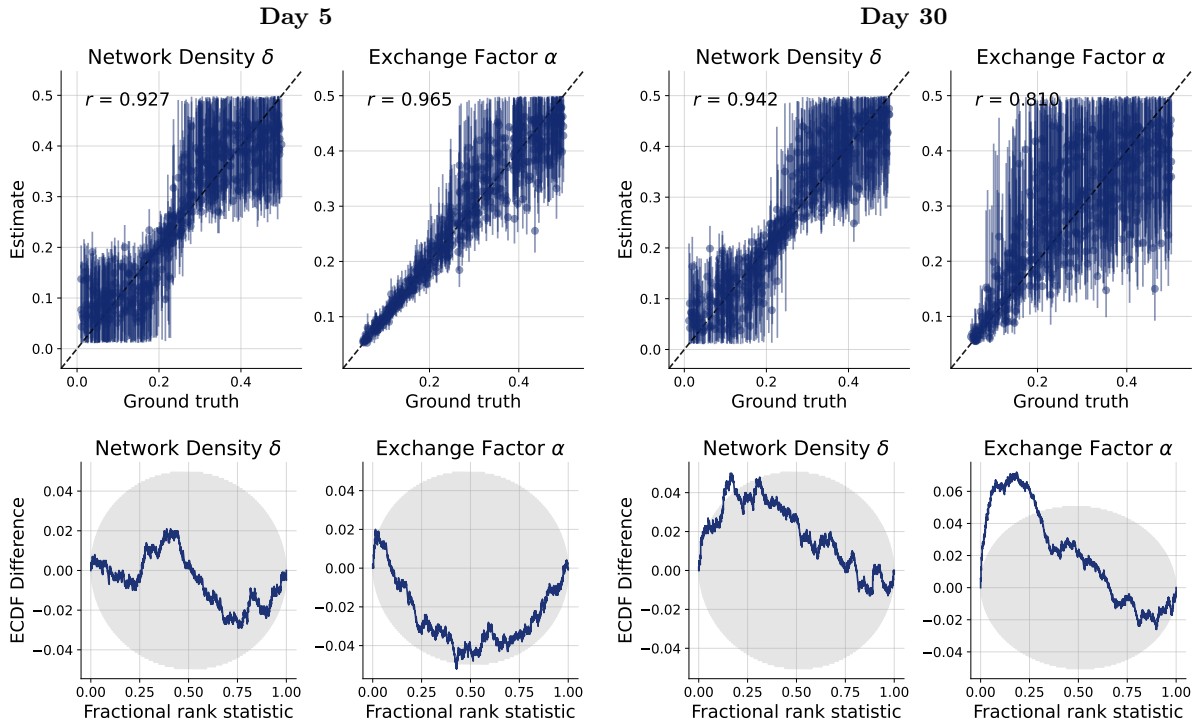

Figure 6: Recovery and calibration of the parameters network density $\delta$ and exchange factor $\alpha$ for each one run of the model using the Set Transformer for a 5-day and 30-day time horizon. The recovery plots show the median value with 95% credible intervals.

## 3.3 Real-world data

To demonstrate the method's applicability to real-world data, we construct a graph using data from Raulo et al. (2023). Social associations between mice were quantified using the adjusted simple ratio index, which measures the frequency with which two mice were observed at the same location within a 12-hour window (see Raulo et al. (2024) for details). Although this spatiotemporal co-occurrence is not a direct measure of social interaction, it is a widely used proxy for social association among individuals (Albery et al., 2021; Sah et al., 2019).

From the microbiome data, we focused on anaerobic non-sporeformers, as these taxa are the primary drivers of microbiome transmission through social networks (Raulo et al., 2024). To keep the analysis computationally feasible, we retained only the taxa with the highest variance in occurrence across mice, excluding taxa that are nearly omnipresent or nearly absent, as these carry little discriminatory information. This resulted in a network of 177 mice and 52 taxa.

Since the graph size differs from those used in previous training, we retrain the model on simulated data that more closely resembles the true dataset. The new training data is generated with adapted hyperparameters and with the observed social association network held fixed. Because the network is fixed, its density is no longer a free parameter, leaving the exchange factor as the sole parameter of interest. For the ABI model, we use a Set Transformer with PMA aggregation as the summary network, motivated by its strong performance in earlier experiments. Since inference is performed on a single parameter, we adopt flow matching (Liu et al., 2022) as the inference network. Both networks are trained together offline on 10 000 simulated datasets.

Figure 7 shows the recovery and calibration evaluated on a simulated test dataset, alongside the posterior density of the exchange factor inferred from the real graph. Recovery on the test data is very high. The calibration shows a tendency toward overestimation, though this may partly be an artifact of the high recovery, as small deviations become detectable when the overall fit is already very good. The posterior density (right panel) is strongly concentrated compared to the uniform prior on (0.05,0.95), with a posterior mean of 0.144 and a posterior standard deviation of 0.009.

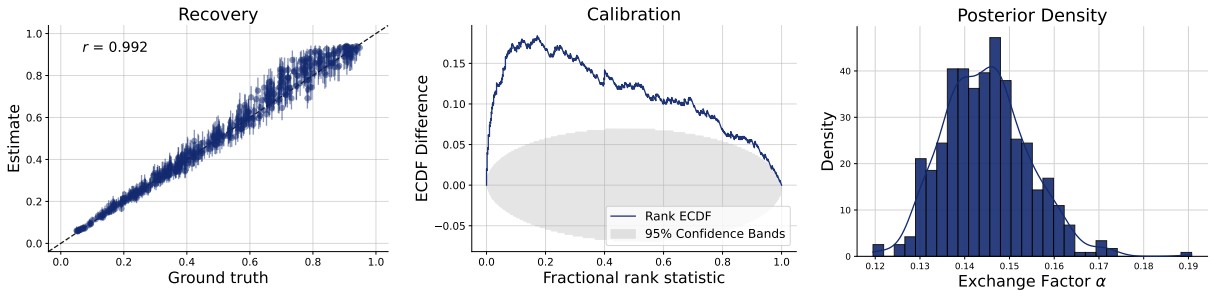

Figure 7: Recovery (left) and calibration (center) of the exchange factor $\alpha$ evaluated on simulated test data for the real-world mice model. The right panel shows the posterior distribution of $\alpha$ inferred from the real-world dataset, displayed as a histogram with a kernel density estimate overlay.

We further evaluate the model using posterior predictive checks. For each posterior draw, we simulate microbiome data using the simulator and summarize the output via the mean and standard deviation of the Jaccard index. The Jaccard index captures the proportion of microbial taxa shared between a pair of mice relative to the total detected across both, providing an intuitive measure of transmission signal (Raulo et al., 2024; 2021).

Figure 8 plots the mean Jaccard index against its standard deviation. The blue points represent prior predictive samples, obtained by simulating data across 100 equally spaced draws from the prior, and thus illustrate the range of summary statistic combinations the simulator can produce. The red points correspond to posterior draws conditioned on the real-world dataset, shown in green. While the mean and standard deviation of the real-world dataset each fall within the prior predictive distribution individually, their combination lies outside the region the simulator can reproduce. Comparing the real-world value to the posterior draws, the standard deviation is well matched, but the mean Jaccard index of the real-world data is substantially higher than that of the posterior simulations.

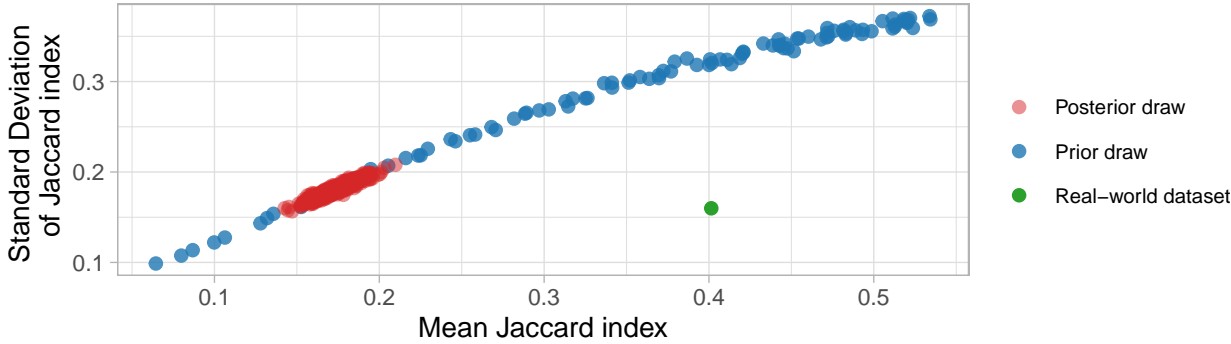

Figure 8: Posterior predictive check for the real-world mouse microbiome dataset. Mean Jaccard index (x-axis) versus standard deviation of the Jaccard index (y-axis) for simulated and observed data. Blue points represent prior predictive samples, illustrating the range of summary statistic combinations attainable by the simulator. Red points correspond to posterior predictive samples conditioned on the real-world dataset. The green point marks the observed value for the real-world dataset.

These results indicate that the model captures the variance in the data reasonably well, but fails to recover the correct location. This discrepancy does not stem solely from the trained model; it primarily reflects a misspecified simulator. The posterior predictive checks reveal a clear simulation gap between the data the simulator can generate and the structure present in the real-world dataset. As this paper focuses on the application of ABI to graph-structured data rather than on capturing the full biological complexity of mouse gut microbiome transmission, we do not attempt to close this gap here.

### 3.4 Train scheduling

Another application domain is logistics, where graph-structured representations arise naturally. Transportation systems consist of interconnected infrastructure and moving entities whose interactions are constrained by this topology, making graphs a principled way to capture capacity, routing, and propagation effects. In particular, rail networks are highly structured, with delays spreading through shared resources and tightly coupled schedules.

In the following experiment we focus on rail traffic. We develop a simulator that mimics a simplified train network and studies how scheduling choices affect the punctuality of a set of trains subject to stochastic, exogenous delays.

#### 3.4.1 Problem setup

We consider a fixed graph with 10 nodes, $V = \{1, 2, \ldots, 10\}$. Each node represents a single track section that can be occupied by at most one train at a time. Node $i$ has an attribute $t_{\mathrm{default}}(i)$, the default traversal time in whole minutes for a train to pass the section in the absence of delays. Trains move through the network by traversing consecutive nodes. In this simplified model, a train is always either traversing a section or waiting on a section to enter the next one; dwell times at stops are assumed negligible.

Our setup includes four trains. Each train starts on a distinct section and follows a predetermined route of four sections to its terminal stop. The primary outcome is the total travel time for each train to complete its four-section route.

In an idealized scenario, the schedule would prevent simultaneous demands for the same section and no exogenous delays would occur. In our simulator, both planned overlaps and stochastic delays are possible. Schedules are randomized, so overlaps may be present even before delays. Additionally, each train incurs an exogenous delay on a section with probability of 10%. Conditional on a delay occurring, the default traversal time of a track section $t_{\mathrm{default}}(i)$ is extended by a random value drawn from a Gamma distribution with shape parameter 5 and rate parameter 0.5. These random delays can create further unplanned overlaps. When two trains request the same section, only one may enter; the other must wait until the section is fully vacated before proceeding. Conflicts are resolved by a waiting-time rule: the train that has been waiting longer for the section goes first; if both have been waiting for the same amount of time, the train with the lower index (e.g., train 1 over train 2) proceeds. This resource-sharing constraint induces strong dependence among the four trains' total travel times.

In our simulation, the track topology is held fixed, whereas both the train schedules and the baseline traversal times of the 10 track sections vary across runs. For each run, the default traversal time of section $i$ is drawn independently as an integer (in minutes) from a discrete uniform distribution over $\{5, 6, \ldots, 25\}$. To generate a schedule, each of the four trains is initialized to depart simultaneously from four distinct sections. Subsequently, at each step a train moves to one of its neighboring sections, chosen uniformly at random, producing randomized routes through the network. Our primary outcomes are the resulting total travel times of the four trains.

#### 3.4.2 Training architecture

To mitigate training instabilities caused by the discreteness of the targets, we add i.i.d. Gaussian noise to the true total travel times: for each train we observe $T^{\mathrm{obs}} = T^{\mathrm{true}} + \epsilon, \epsilon \sim \mathcal{N}(0, 1)$. The summary network receives, as input, a $10 \times 17$ matrix: each row corresponds to one of the 10 sections. The first column contains $t_{\mathrm{default}}(i)$. The remaining 16 columns provide a one-hot encoding of the schedule/route assignments across

trains and positions: there is one binary indicator for each (train, step) pair — four trains × four steps — set to 1 in the row of a section if that section is used by that train at that step, and 0 otherwise. Because the graph structure is fixed and known, we do not supply an adjacency matrix.

Since the Set Transformer performed best in the two previous experiments, we adopt it here as the summary network, using multi-head attention pooling. Because this experiment targets neural likelihood estimation rather than recovery of low-dimensional parameters, we increase the summary dimension to 64. As before, the inference network is implemented as a coupling flow with spline-based transformations. The training dataset comprises 10 000 distinct combinations of train schedules and baseline traversal times. Each combination is observed 64 times, yielding a total of 640 000 training samples.

### 3.4.3 Results

Figure 9 summarizes posterior recovery and calibration for all four trains. The top row reports recovery of the posterior medians, showing correlations above 0.88 for every train, which indicates that the central tendency of the posterior is well captured. The vertical lines denote the 95% credible intervals; the posterior medians are often shifted away from their centers, suggesting skewed posterior distributions. The bottom row presents ranked ECDF calibration plots. Overall, calibration is good for all trains, although Trains 2 and 3 exhibit slight underestimation in the lower value range.

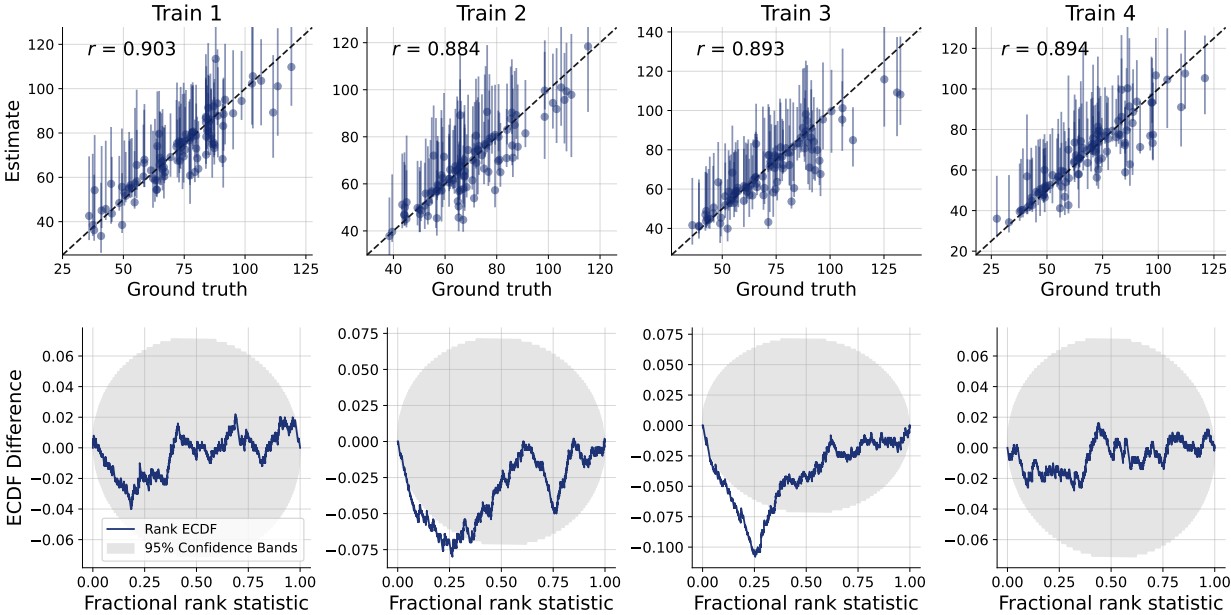

Figure 9: Recovery plots (on the top) and calibration ECDF plots (on the bottom) for the total travel time of the four trains. Vertical lines in the recovery plots correspond to 95% credible intervals.

In Figure 10 we present results for two combinations of schedules and baseline traversal times, with each row corresponding to one specific setup. Ground-truth densities are approximated by running the simulator 500 times per setup and applying Gaussian kernel density estimation to the resulting total travel times. The estimated posteriors are based on 500 draws from the learned posterior distribution. The ground-truth densities show that nearly all total travel-time distributions are right-skewed and often multimodal. The dominant peak corresponds to the travel time achieved when no random delays occur and the schedule can be executed as planned — even in the presence of double-booked tracks, where one train must wait for another to clear the section. Overall, the estimated posterior densities closely match the ground-truth distributions.

As an example, in the first setup (first row of Figure 10), Train 1 exhibits a bimodal distribution, with the smaller mode occurring less frequently. This pattern is explained by planned schedule overlaps: Train 1 typically needs to wait for another train to clear a contested track section. It can proceed immediately only when the competing train experiences a random delay on an upstream section, allowing Train 1 to pass first. An analogous bimodal pattern for Train 1 appears in the second setup (second row) for the same reason.

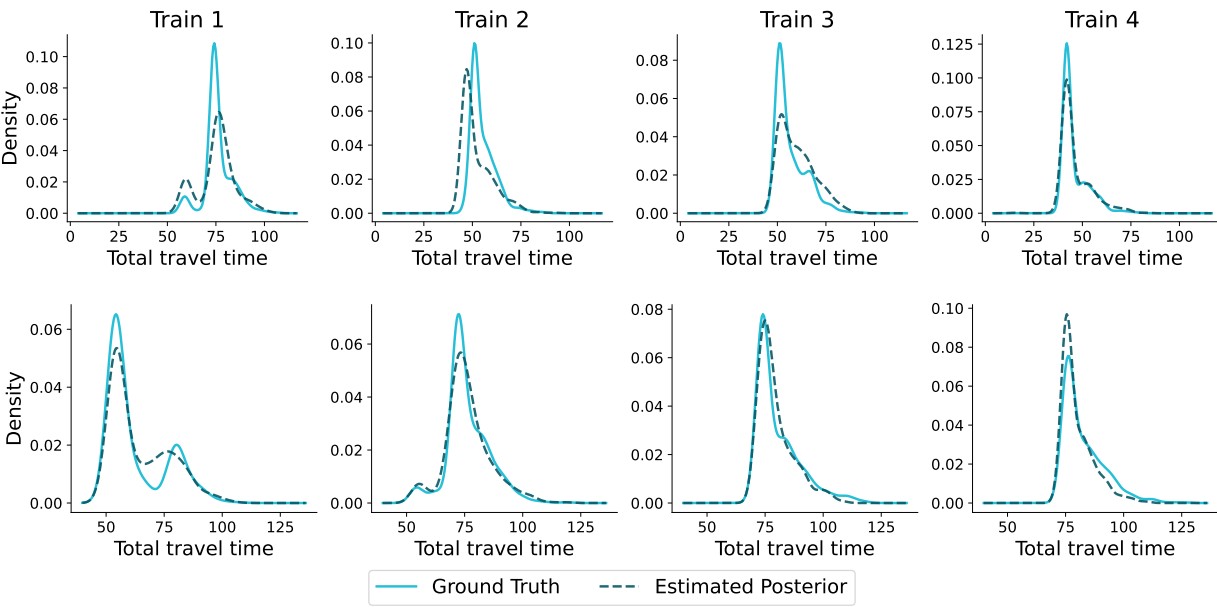

Figure 10: Two example settings (one per row) showing the estimated posterior densities of total travel time for the four trains, alongside the corresponding ground-truth densities. Ground truth is approximated by 500 simulator runs per setting, using the resulting travel-time samples. All densities are computed from samples via Gaussian kernel density estimation.

## 4 Summary

The paper extends Amortized Bayesian Inference (ABI) to graph-structured data by pairing a permutation-invariant summary network with a flexible neural inference network for fast likelihood-free posterior estimation. The key methodological challenge is that graph summaries must remain invariant to node relabeling in the graph, work across varying sparsities and sizes, and capture long-range dependencies that local message passing often misses. To tackle these issues, the paper introduces and systematically evaluates several graph-aware summary networks across multiple graph-based inference tasks. Specifically, it compares a Graph Convolutional Network, a Graph Transformer, and a Set Transformer against a Deep Sets baseline, each paired with different pooling mechanisms. Posterior quality is assessed not only by parameter recovery but also by simulation-based calibration and posterior contraction, emphasizing trustworthy uncertainty quantification.

Across three empirical case studies, the paper evaluates graph-aware ABI using several permutation-invariant summary networks. In the first case study, a controlled toy simulator generates undirected graphs with type-specific baseline edge probabilities and an additional triadic-closure parameter. Most architectures infer the baseline probabilities accurately, yielding high parameter recovery and strong posterior contraction. In contrast, the triadic-closure parameter reflected higher-order, local structure, making it substantially harder to identify; among the tested models, the Deep Sets, Graph Transformer and Set Transformer achieve high parameter recovery and posterior contraction. Achieving good calibration remains challenging for architectures, which achieve good parameter recovery. Second, in a biological simulator of weighted social interaction networks among mice driving microbiome exchange, the global network density is inferred reliably but was

not well calibrated, while the exchange factor was more difficult. Across all considered summary-network architectures, parameter recovery declines as the observation horizon increases, consistent with reduced identifiability once the microbiome dynamics approach a steady state. The Set Transformer again outperforms the other three architectures in terms of recovery and posterior contraction. Interestingly, the best calibration for the network-density parameter is achieved by Deep Sets and the Graph Transformer, and for the exchange-factor parameter by the Graph Convolutional Network, even though the Graph Convolutional Network performs worst overall in recovery and contraction. Third, in a train-scheduling simulator defined on a fixed rail-section graph, the task is to perform neural likelihood estimation for the trains' total travel times. Here, the Set Transformer combined with attention pooling produced well-calibrated posteriors over the full travel-time distributions: posterior medians closely track the ground truth, and the inferred densities capture the characteristic right-skewness and frequent multimodality induced by stochastic delays and traffic conflicts.

Overall, the study shows that ABI can be made graph-aware and practically useful for graph parameter inference, provided the summary network is expressive enough. The results suggest the Set Transformer with global attention and a multi-head attention pooling layer as a strong default for ABI on graphs, especially when long-range dependencies matter. By contrast, the Graph Convolutional Network is not expressive enough to achieve good parameter recovery and posterior contraction in the settings considered, despite explicitly exploiting graph structure via convolutional layers and $k$-hop neighborhoods. Similarly, the Graph Transformer — an adaptation of the Set Transformer that also incorporates explicit graph structure — does not outperform the standard Set Transformer architecture. This is somewhat surprising: the Set Transformer receives only generic node features and must implicitly learn how to interpret both the adjacency information and node attributes, whereas the Graph Convolutional Network and Graph Transformer are architecturally designed to process and exploit graph structure directly.

A key limitation of the work is the narrow range of graph types and scales considered. All three case studies involve relatively small graphs with fewer than 50 nodes. In many real-world domains, graphs routinely contain more than $10^5$ nodes, introducing fundamentally different challenges in data handling, memory, and model scalability. Similar issues arise from extremely sparse networks or graphs with heavy-tailed degree distributions. Moreover, the paper focuses exclusively on undirected graphs. Extending the approach to directed graphs, temporal or dynamic graphs, and heterogeneous graphs with multiple node or edge types would likely require additional architectural and methodological adjustments, and remains an important direction of future research.

### Acknowledgments

This research was supported by the German Research Foundation (DFG) via Collaborative Research Cluster 391 "Spatio-Temporal Statistics for the Transition of Energy and Transport" – 520388526 and Project 528702768.

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

# A   Toy Example: Additional Results

Table 2: Results for the toy example across four summary-network architectures, each evaluated with three pooling variants. We report the loss value at epoch 250 (which is comparable between architectures, since the same inference network is used throughout), the parameter recovery (PR), calibration ($\ell_\gamma$) and the posterior contraction (PC) for $\bar{\pi}$ as the average of the results of $\{\pi_{AA}, \pi_{BB}, \pi_{AB}\}$ and the parameter $\lambda$, as well as the number of parameters used in the summary network (# pars).

| Summary Netw. | Aggregation | # pars | Loss | PR $\bar{\pi}$ | PR $\lambda$ | $\ell_\gamma\bar{\pi}$ | $\ell_\gamma\lambda$ | PC $\bar{\pi}$ | PC $\lambda$ |
|---|---|---|---|---|---|---|---|---|---|
| | Mean | $3.2 \cdot 10^5$ | 1.84 | 0.80 | 0.80 | -5.12 | 0.49 | 0.67 | 0.62 |
| DeepSets | Invariant | $3.3 \cdot 10^5$ | 1.70 | 0.82 | 0.85 | -3.03 | -9.94 | 0.72 | 0.72 |
| | PMA | $4.8 \cdot 10^5$ | 1.84 | 0.80 | 0.80 | -1.40 | 2.42 | 0.66 | 0.61 |
| | Mean | $2.4 \cdot 10^5$ | 5.59 | 0.15 | 0.23 | 0.21 | 0.58 | 0.00 | 0.00 |
| GCN | Invariant | $2.4 \cdot 10^5$ | 5.58 | 0.31 | 0.64 | 1.67 | -9.85 | 0.04 | 0.27 |
| | PMA | $4.0 \cdot 10^5$ | 3.61 | 0.67 | 0.76 | 3.16 | 1.19 | 0.36 | 0.54 |
| | Mean | $3.1 \cdot 10^5$ | 2.63 | 0.80 | 0.75 | 0.46 | -15.51 | 0.64 | 0.61 |
| GraphTransformer | Invariant | $3.3 \cdot 10^5$ | 2.59 | 0.80 | 0.75 | 2.20 | -26.08 | 0.65 | 0.64 |
| | PMA | $4.6 \cdot 10^5$ | 2.82 | 0.80 | 0.75 | 0.08 | -23.19 | 0.59 | 0.61 |
| | Mean | $4.1 \cdot 10^5$ | 1.62 | 0.83 | 0.86 | 1.72 | -5.80 | 0.71 | 0.73 |
| SetTransformer | Invariant | $4.4 \cdot 10^5$ | 1.62 | 0.83 | 0.86 | 0.09 | -4.60 | 0.74 | 0.76 |
| | PMA | $5.7 \cdot 10^5$ | 1.59 | 0.83 | 0.86 | 1.86 | -6.30 | 0.72 | 0.73 |

Table 3: Training time in minutes for different training dataset sizes across the four summary network architectures in the toy example from Section 3.1. All networks use the invariant aggregation layer (Deep Sets and GCN) or PMA aggregation (Graph Transformer and Set Transformer). Once trained, all architectures generate posterior samples near-instantaneously, requiring approximately 0.1 seconds for 1000 draws. All reported runtimes were measured on an Intel Core Ultra 9 185H CPU (2.30 GHz) with 32 GB of RAM.

| Training Data Size | DeepSets | GCN | GraphTransformer | SetTransformer |
|---|---|---|---|---|
| 3,200 | 4.94 | 3.38 | 9.04 | 13.78 |
| 6,400 | 10.59 | 6.41 | 29.47 | 50.67 |
| 12,800 | 36.40 | 11.90 | 58.40 | 73.95 |
| 32,000 | 127.88 | 41.10 | 175.56 | 294.58 |

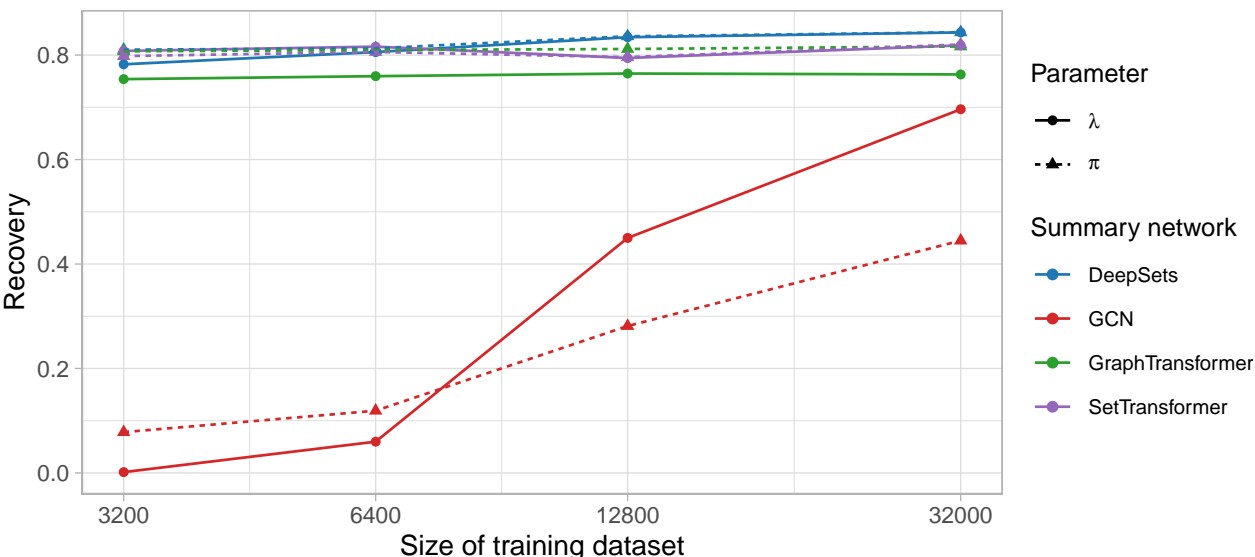

Figure 11: Recovery of the parameters $\pi = \{\pi_{AA}, \pi_{BB}, \pi_{AB}\}$ and $\lambda$ for four summary network architectures as a function of training dataset size. All networks use either the invariant aggregation layer (Deep Sets and GCN) or PMA aggregation (Graph Transformer and Set Transformer). For all architectures except the GCN, recovery improves only marginally with increasing dataset size, whereas the GCN shows substantial gains as more training data becomes available.

## B  Toy Example: Data-dependent SBC

In the main text, we discussed the performance of the networks measured with SBC on model parameters, but calibration can also be evaluated at the level of the data itself. This approach can be understood as a form of posterior predictive checking; since we operate in a simulation-based setting, we follow Modrák et al. (2025) and employ data-dependent test quantities for SBC. A natural and well-established choice is the model likelihood, but in many settings where SBI is applied, the likelihood is intractable and therefore unavailable as a test quantity. Since our data are graphs, we require test quantities that summarize graph structure. We opt for four such measures: the spectral gap, edge density, degree assortativity, and the global clustering coefficient.

The spectral gap is the difference between the first and second eigenvalue of the normalized graph Laplacian. A larger spectral gap indicates a well-connected graph that is difficult to disconnect, whereas a small spectral gap is characteristic of graphs with pronounced community structure or near-disconnection (Fiedler, 1973).

Edge density is the fraction of possible edges that are present in the graph. It provides a basic summary of graph sparsity and directly corresponds to the connection probability in Erdős–Rényi random graphs (Newman, 2010).

Degree assortativity is the Pearson correlation coefficient between the degrees of adjacent node pairs. Positive values indicate that high-degree nodes tend to connect to other high-degree nodes, a pattern commonly observed in social networks, while negative values indicate that hubs tend to connect to low-degree nodes, as is typical in biological and technological networks (Newman, 2002).

The global clustering coefficient, also referred to as transitivity, measures the fraction of connected node triplets that close into triangles: $3 \times \text{triangles}/\text{triplets}$. It captures the tendency of a network to form tightly knit local groups (Watts & Strogatz, 1998).

To obtain values of the introduced metrics for a fitted model, the following steps are performed: A graph sampled from the simulator serves as the test graph, and posterior draws of the parameters are obtained using the fitted model. Each posterior draw is then passed into the simulator to generate a new graph, for

which the metrics described above are computed and compared to those calculated from the original test graph.

Figure 12 shows the results of the data-dependent SBC for the toy example from Section 3.1, covering all twelve summary networks and the four data-dependent test metrics. The left column displays the recovery, i.e., the correlation between the true metric value and the median of the metrics computed from the posterior draws. For the Deep Sets, Graph Transformer, and Set Transformer architectures, recovery is very high across all four data-dependent metrics. In particular, the Deep Sets and Set Transformer achieve a recovery of 1 for both edge density and global clustering coefficient. The GCN struggles to attain good recovery when using mean or invariant aggregation, whereas the GCN with PMA aggregation performs comparably to the other three network types. Calibration of these metrics is challenging, particularly when recovery is very high, as even minor miscalibration becomes detectable. For edge density, the Deep Sets and Set Transformer models achieve near-perfect recovery but fail to calibrate well — the values of $\ell_\gamma$ for the Set Transformer collapse to $-\infty$. For degree assortativity, all networks achieve good calibration, with $\ell_\gamma > 0$ throughout.

Overall, most networks show satisfactory SBC performance with the chosen data-dependent test quantities. The calibration difficulties can be attributed to the extremely high recovery of the metrics and should therefore not be interpreted as evidence of poor calibration.

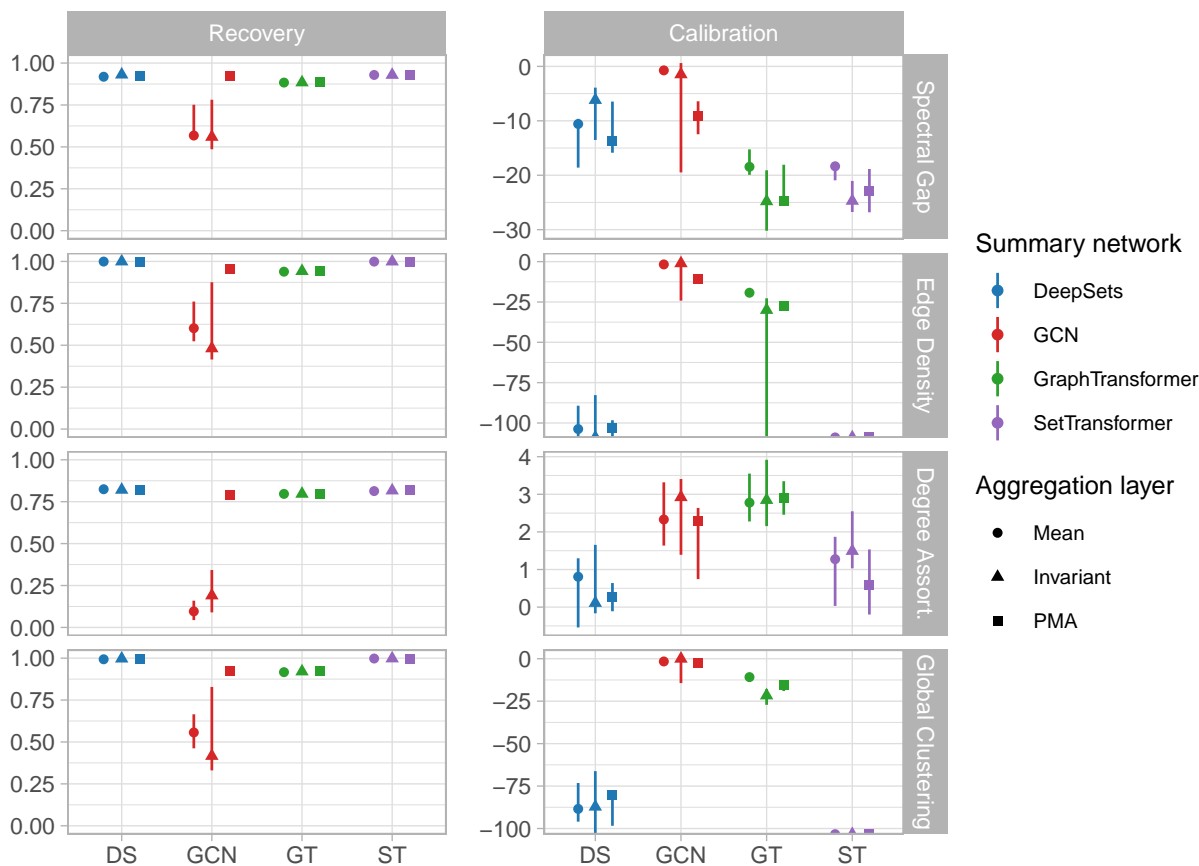

Figure 12: Results for the toy example across four summary-network architectures with each three different aggregation layers. We report recovery (higher is better) and calibration ($\ell_\gamma$; values $> 0$ indicate good calibration) for the metrics spectral gap, edge density, degree assortativity and the global clustering coefficient. Markers (circles/triangles/squares) indicate the median across five runs, and error bars span the minimum to maximum values. Markes at the very bottom of the plot represent the value of $-\infty$.

## C   Toy example: Comparison to MCMC baseline

To compare our proposed approach against an MCMC baseline, a tractable likelihood is required. However, in all three experiments — including the toy example — this is not the case. To nonetheless provide an indication of how the ABI approach compares to MCMC, we introduce a simplified version of the toy example in which only the parameters $\pi_{AA}$, $\pi_{BB}$, and $\pi_{AB}$ are used to simulate a graph. The triadic closure parameter $\lambda$ is removed to yield a tractable likelihood.

We train a model on this reduced simulator using a Set Transformer with PMA aggregation as the summary network and a coupling flow with spline transformation as the inference network within BayesFlow (Kühmichel et al., 2026). As a baseline, we fit the same model via MCMC, specifically HMC (Neal, 2011) implemented in Stan (Stan Development Team, 2024). Online training of the BayesFlow model for 150 epochs with 100 batches of size 32 per epoch (480,000 datasets in total) took approximately 38.85 minutes. Drawing 500 posterior samples for each of 500 test datasets was then nearly instantaneous, requiring only 0.92 seconds in total. For the MCMC baseline, four separate chains are run for each of the 500 test graphs, which took 33.43 minutes in total.

Figure 13 shows calibration ECDF plots for both the BayesFlow model and the MCMC baseline. Both approaches achieve good calibration for $\pi_{AA}$, $\pi_{BB}$, and $\pi_{AB}$. The log-likelihood, however, is only well calibrated under MCMC; the BayesFlow model struggles to reach comparable calibration for this quantity. This discrepancy is expected, as the BayesFlow networks are trained using a simulation-based loss rather than the likelihood-based objective that underpins MCMC.

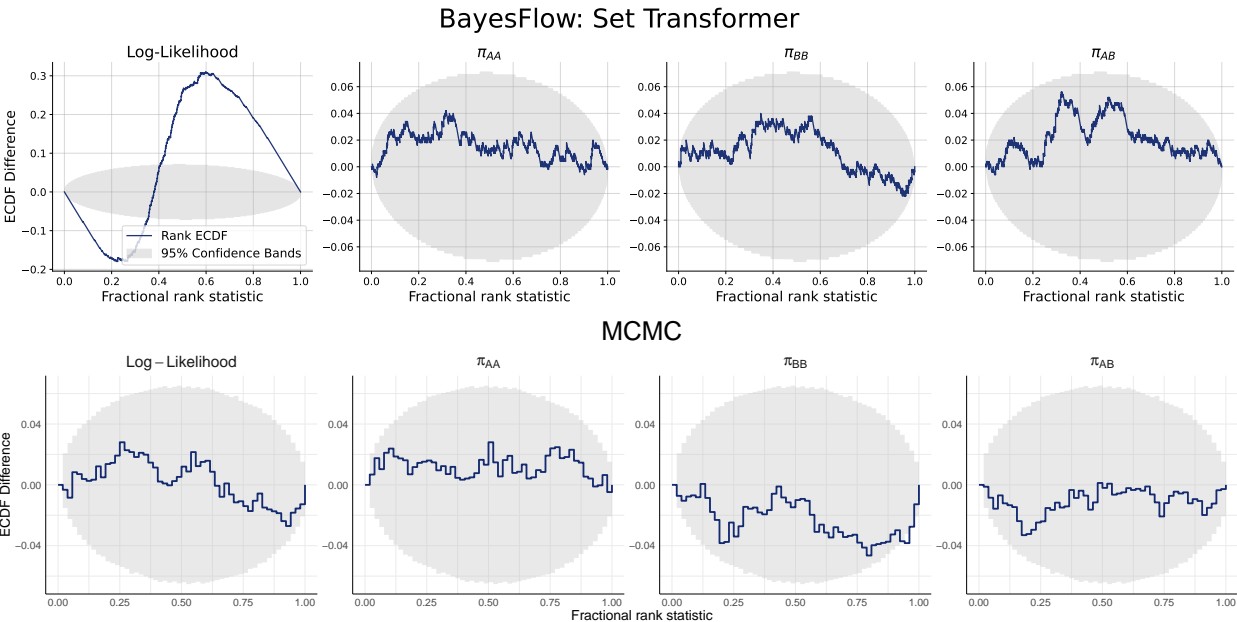

Figure 13: Calibration ECDF plots for the BayesFlow model using a Set Transformer (with PMA aggregation) as the summary network compared to a MCMC baseline in the simplified version of the toy experiment. In addition to the parameters of the simulator $\pi_{AA}, \pi_{BB}$ and $\pi_{AB}$, the calibration of the log-likelihood is shown.

Figure 14 shows the recovery of the three parameters, with MCMC posterior medians on the x-axis and the corresponding BayesFlow posterior medians on the y-axis. Error bars represent 95% credible intervals. The posterior medians from BayesFlow closely match those from MCMC, and the widths of the error bars are similar in both directions, indicating that the two approaches yield comparable posterior uncertainty.

Taken together, these results demonstrate that the BayesFlow model closely approximates the MCMC baseline. It should be noted, however, that MCMC is only applicable when an analytical likelihood is available,

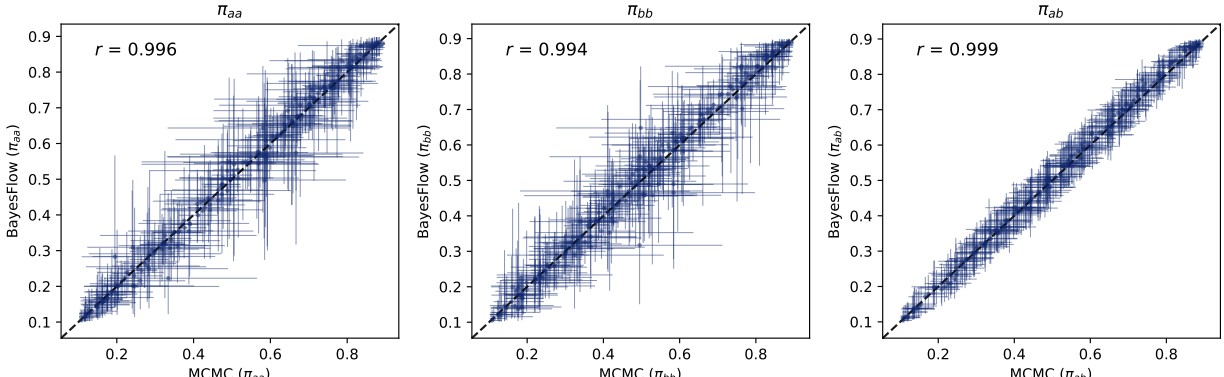

Figure 14: Recovery of the parameters $\pi_{AA}, \pi_{BB}$ and $\pi_{AB}$ for the MCMC baseline vs the model fitted with BayesFlow (summary network: Set Transformer with PMA aggregation; inference network: coupling flow with spline transformation). The horizontal and vertical lines correspond to the 95 % credible intervals.

which was not the case in the experiments presented in Section 3. Furthermore, while the total compute times are comparable — 38 minutes of training for BayesFlow versus 33 minutes for MCMC across 500 test graphs — the BayesFlow model has a decisive practical advantage: once trained, it produces posterior samples for any new graph in under a second, whereas MCMC must be run from scratch for each new observation.

## D    Mice interaction network: Additional Results

Table 4: Results for the mice interaction case study across four summary-network architectures and three different observations horizons (5, 10 and 30 days). The validation loss (Val. loss) is measured at the last epoch, before the training stopped due to early stopping (the maximal number of 250 epochs was not reached in any training run). We report parameter recovery (PR), calibration ($\ell_\gamma$), and posterior contraction (PC) for both parameters: network density ($\delta$) and exchange factor ($\alpha$).

| Day | Summary Network | Val. loss | PR $\delta$ | PR $\alpha$ | $\ell_\gamma$ $\delta$ | $\ell_\gamma$ $\alpha$ | PC $\delta$ | PC $\alpha$ |
|---|---|---|---|---|---|---|---|---|
| 5 | DeepSets | 0.98 | 0.93 | 0.87 | -1.47 | -15.12 | 0.88 | 0.79 |
| | GCN | 2.13 | 0.86 | 0.37 | -7.17 | 0.38 | 0.76 | 0.11 |
| | GraphTransformer | 0.99 | 0.91 | 0.91 | -1.26 | -14.89 | 0.84 | 0.84 |
| | SetTransformer | 0.30 | 0.94 | 0.97 | -3.03 | -8.98 | 0.88 | 0.95 |
| 10 | DeepSets | 1.26 | 0.93 | 0.79 | -1.68 | -2.42 | 0.88 | 0.64 |
| | GCN | 2.22 | 0.84 | 0.33 | -11.91 | 0.94 | 0.73 | 0.12 |
| | GraphTransformer | 1.25 | 0.90 | 0.83 | -1.79 | 0.95 | 0.82 | 0.69 |
| | SetTransformer | 0.67 | 0.93 | 0.91 | -8.49 | -4.19 | 0.89 | 0.86 |
| 30 | DeepSets | 1.46 | 0.94 | 0.68 | -1.49 | -7.67 | 0.88 | 0.50 |
| | GCN | 3.21 | 0.74 | 0.20 | $-$ inf | -2.19 | 0.59 | 0.08 |
| | GraphTransformer | 1.93 | 0.85 | 0.72 | -1.75 | -24.45 | 0.74 | 0.54 |
| | SetTransformer | 1.22 | 0.94 | 0.82 | -2.13 | -12.04 | 0.90 | 0.70 |

