# OpenReview forum: "From Mice to Trains:  Amortized Bayesian Inference on Graph Data"
_TMLR — Accepted by TMLR_

### Review · Reviewer_R9XX · 2026-01-23

**Summary Of Contributions:**

The paper proposes an amortised bayesian inference (ABI) method for graph data by adapting the two-stage ABI method of Radev (2023). The paper compares four architectures for processing the graph data, and applies the method two real world use cases.

**Audience:**

Yes

**Audience Explanation:**

The research topic is timely, significant and interesting for ML audience.

**Broader Impact Concerns:**

No issues.

**Claims And Evidence:**

No

**Claims Explanation:**

The comparison setting of the paper is quite light. There is only a single simple synthetic toy case where the four variants of the method are compared against. There are no comparisons to any other methods, and the experiment is performed using only (I assume) 3200 graphs.

The paper should improve the empirical evaluation on all three accounts:
- Evaluating the method using only a single binary network is not sufficient. The paper should consider some synthetic experiment more similar to real world use cases.
- Some competing methods should be added, which should cover both simple baselines and state of the art. Here prior fitted networks seem like an apt comparison target
- It would be very useful to repeat the experiments using varying number of simulated datasets, which would give performance curves of the methods wrt the size of simulations.

In the mice experiment the paper should demonstrate more clearly why the simulations are relevant for the real mice problem. ABI depends on having a simulation design that matches the real-world process, and here it seems that the simulation is quite simplistic, and likely does not capture the complexity of real mice behavior. The authors should discuss and analyse this further.

**Requested Changes:**

See above.

---

> ### Author Response · Authors · 2026-02-24
>
> Thank you for your feedback. Here are our answeres to your main concerns:
>
> ## Real-world use cases of ABI
> Our method was evaluated not only in the synthetic toy case study, but also in two experiments involving more realistic, real-world problems. In the mice interaction network experiment, we additionally compare across different summary network architectures.
>
> To further demonstrate the practical relevance of this case study, we now incorporate in the revised version real observational data from [CEH Data Catalogue](https://catalogue.ceh.ac.uk/documents/043513e5-406c-4477-89aa-c96059acb232), focusing on the social interaction network among mice. Specifically, we consider anaerobic non-spore-formers, as these taxa are known to spread between mice through direct social contact (see [Raulo et al., 2024](https://www.nature.com/articles/s41559-024-02381-0)). Due to the larger scale of the real dataset  (177 mice and 52 taxa variables) we retrain the model on newly simulated data with a fixed ground truth graph, performing inference on the exchange factor parameter only. The resulting posterior distributions and posterior predictive checks support the validity and meaningfulness of the modelling approach. A detailed analysis will be added to Section 3.2.
>
> ## Comparison to MCMC baseline
> Comparing our ABI method to a baseline such as MCMC is challenging because MCMC requires a tractable likelihood. In both real-world experiments — the mice interaction network and the train scheduling problem — this requirement cannot be met, as the likelihood is intractable. Even the toy example we constructed has an intractable likelihood due to the triadic closure parameter $\lambda$. A direct comparison to MCMC or similar methods is therefore not feasible in these settings.
>
> We will include a section in the appendix, where the toy case study is reduced to the parameters $\pi_{AA}, \pi_{BB}$ and $\pi_{AB}$, by removing the triadic closure parameter $\lambda$. This makes the likelihood tractable and therefore the calculation of the problem using MCMC as a baseline feasible.
>
> For this comparison, we train a BayesFlow model using the Set Transformer as the summary network. After training, we compare the resulting posterior distributions to those obtained via MCMC (specifically, adaptive HMC implemented in Stan). The posterior means are in close agreement across both methods, with similarly sized 95% credible intervals. MCMC achieves satisfactory calibration for all parameters and the log-likelihood, while the BayesFlow model is well-calibrated for the parameters but shows greater difficulty in calibrating the log-likelihood. This discrepancy is expected, as the network is trained using a simulation-based loss rather than the likelihood-based objective that underpins MCMC. We will add more detailed results including plots into a section of the appendix.
>
> ## Comparison to PFNs
> Regarding prior-fitted networks (PFNs), these are not applicable to our setting. PFNs are designed to approximate the posterior predictive distribution over new observations, whereas our interest lies in the posterior over parameters or latent variables. To our knowledge, no existing method of this type is capable of performing the inference we require.

---

> > ### Author Response · Authors · 2026-02-24
> >
> > ## Performance for varying number of datasets
> > Investigating how the performance of our method scales with training data size is a valuable analysis. To this end, we focus on the first case study from Section 3.1 and train the DeepSet, GCN, Graph Transformer, and Set Transformer on datasets of size 3200, 6400, 12800, and 32000. The recovery of $\pi_{AA}, \pi_{BB}$ and $\pi_{AB}$ is reported in the first table below, followed by the recovery of $\lambda$ in the second table. We will add corresponding results to the appendix.
> >
> > Table 1: Mean recovery of the parameters $\pi_{AA}, \pi_{BB}$ and $\pi_{AB}$.
> > | Dataset Size | Deep Sets | GCN | Graph Transformer | Set Transformer|
> > |-------------:|-----------------------:|-------------------:|-----------------------:|---------------------:|
> > | 3200         | 0.779                  | 0.019              | 0.772                  | 0.766                |
> > | 6400         | 0.814                  | 0.030              | 0.778                  | 0.776                |
> > | 12800        | 0.823                  | 0.220              | 0.781                  | 0.776                |
> > | 32000        | 0.819                  | 0.466              | 0.783                  | 0.799                |
> >
> > Table 2: Recovery of the parameter $\lambda$.
> > | Dataset Size | Deep Sets | GCN | Graph Transformer | Set Transformer |
> > |-------------:|-----------------------:|-------------------:|-----------------------:|---------------------:|
> > | 3200         | 0.806                  | -0.024             | 0.772                  | 0.814                |
> > | 6400         | 0.833                  | 0.134              | 0.774                  | 0.834                |
> > | 12800        | 0.860                  | 0.484              | 0.790                 | 0.824                |
> > | 32000        | 0.860                  | 0.680              | 0.796                 | 0.845                |

---

### Review · Reviewer_FbSe · 2026-02-07

**Summary Of Contributions:**

This work explores a range of neural architectures for the encoding of graph structured data in the context of amortized bayesian inference. The paper contains a nice introduction to ABI which is helpful for people getting into the field. The modeling choices considered for graph encoding are diverse and some very reasonable ones have been selected for experiments. Experiments include a toy dataset and two real-world datasets, with adequate analysis into the results.

**Additional Comments:**

I understand the methodologies used by this paper, but I haven't been closely following the field of amortized bayesian inference. For example, is the encoding aspect the bottleneck in ABI? I fear not. The simulation method or inference network may be more crucial for the posterior sample quality. I recommend the editors to deter their decisions to opinions more authoritative in this area of research.

**Audience:**

Yes

**Audience Explanation:**

For people interested in seeing the performance of different graph encoding methods in the context of amortized bayesian inference.

**Broader Impact Concerns:**

No concerns in that regard.

**Claims And Evidence:**

Yes

**Claims Explanation:**

The claims made in this paper is very limited — they specifically focused on the graph encoding aspect of ABI and all experiments are done surrounding that. This is an entirely empirical work which in my opinion is essentially an extended ablation study. But overall the experimental results presented in this work are good evidence to support their research goal and conclusions.

**Requested Changes:**

While the comparison of different graph encoding methods are helpful, it would be really nice to have some other baselines to establish an expectation in reasonable performance in the two real-world datasets used by this paper. MCMC is an example, which is a good postive control choice as I expect that it should give very good results. More contemporary choices include variational methods, generative flow networks and etc. Note that I have limited experience in the ABI field before so I can't suggest very specific baseline methods for comparison.

---

> ### Author Response · Authors · 2026-02-24
>
> Thank you for your valuable feedback.
>
> ## Comparison to MCMC baseline
> Comparing our ABI method to a baseline such as MCMC is challenging because MCMC requires a tractable likelihood. In both real-world experiments (the mice interaction network and the train scheduling problem) this requirement cannot be met, as the likelihood is intractable. Even the toy example we constructed has an intractable likelihood due to the triadic closure parameter $\lambda$. A direct comparison to MCMC or similar methods is therefore not feasible in these settings.
>
> We will include a section in the appendix, where the toy case study is reduced to the parameters $\pi_{AA}, \pi_{BB}$ and $\pi_{AB}$, by removing the triadic closure parameter $\lambda$. This makes the likelihood tractable and therefore the calculation of the problem using MCMC as a baseline feasible.
>
> For this comparison, we train a BayesFlow model using the Set Transformer as the summary network. After training, we compare the resulting posterior distributions to those obtained via MCMC (specifically, adaptive HMC implemented in Stan). The posterior means are in close agreement across both methods, with similarly sized 95% credible intervals. MCMC achieves satisfactory calibration for all parameters and the log-likelihood, while the BayesFlow model is well-calibrated for the parameters but shows greater difficulty in calibrating the log-likelihood. This discrepancy is expected, as the network is trained using a simulation-based loss rather than the likelihood-based objective that underpins MCMC. We will add more detailed results including plots into a section of the appendix.
>
> ## Relevance of the summary network in ABI
> In ABI, the summary network is often the primary bottleneck for posterior quality. The simulator is determined by the scientific problem at hand and cannot be considered a bottleneck in the context of posterior inference; while the choice of simulator can give rise to issues such as the simulation gap or model misspecification, these are outside the scope of the present work. The inference network can become a bottleneck when the parameter posteriors are multimodal or high-dimensional, for instance, involving hundreds of parameters ([Radev et al., 2023](https://proceedings.mlr.press/v216/radev23a.html)), but our experiments do not present these challenges. The summary network is therefore the component that most critically determines posterior sample quality in our setting. Obtaining informative and ideally sufficient summary statistics is a prerequisite for accurate posterior inference, particularly when the observed data have complex structure such as graphs. Our experiments demonstrate that the choice of summary network has a meaningful impact on posterior quality and calibration in this setting, and we believe this constitutes a contribution of independent interest to practitioners working with graph-structured data.

---

### Review · Reviewer_MibK · 2026-02-17

**Summary Of Contributions:**

The paper presents a study of applying amortized bayesian inference (ABI) to graph data, extening in particular  setup from the cited Radev et al. to to use permutation invariant networks to generate a summary vector s which is then mapped onto a gaussian sample z \sim \mathcal{N}(0,I) via a coupling flow (conditional normalizing flow) using spline transformations. The authors study four summary network architectures  (deep sets, graph convolutional networks, graph transformer, set transformer) on a toy problem(estimating probabilities of node connections in a 2 community graph with addiitonal triadic closures) and two real world setups (social interactions between free ranging mice and their relationship to gut microbiome and a train scheduling setup modeling total travel times for four trains between 10 nodes). Surprisingly, set transformers outperform the other architectures , despite not having any particular graph bias.

Overlal, the method shows good recovery and calibration on the toy examples baseline probablities, meaning only the transformer+set based methods perform well. For the mice case study, the architectures showd good recovery but struggled with calibration on network density, while exchange factors were more challenging, with the set transformer overall performing better. An interesting note was that the GCN was best calibrated for the exchange factor despite performing worst in recovery and contraction of the posterior.

Finally, on the train dataset, the set trassformer performs strongly with well calibratated poseriors closely tracking ground truth via medians.

Authors acknoweldge that the study studies tiny undirected networks that do not exhibit the heavy tailed degree distribution or scaling issues of real world networks, as well as not studying the effects directedness,temporality and heterogeneity.

**Audience:**

Yes

**Audience Explanation:**

I was not familiar with SBI and ABI myself before taking on this review and found it interesting to learn about through this paper, as was to the fact that at least for the problems studied, the graph structure does not appear to matter. I'd be interested in further studies where the structure becomes more important for efficiency or inductive bias.

**Broader Impact Concerns:**

Not required

**Claims And Evidence:**

Yes

**Claims Explanation:**

The claims are carefully scoped and well supported by the experiment, being mainly an exploration of the feaisbility of graph ABI.

Based on my reading of  https://arxiv.org/abs/2110.06581   and https://arxiv.org/abs/2101.04653 as well as other SBI work that I did for this review, the authors cover the key modes of evaluation of SBI methods (coverage, calibration, recovery) in what seems like a decently rigorous setup for the scale considered.  The only missing element is posterior predictive coverage, see requested changes.  Code is available and seems reasonable on a quick scan.

**Requested Changes:**

# Nice to have

I was not familiar with SBI before and went on a reading binge and found what I think might be some missing references? feel free to ignore these if not relevant

- https://arxiv.org/abs/2110.06581 and https://arxiv.org/abs/2101.04653 for limitations and considerations on how to benchmark
- https://arxiv.org/abs/2304.06806 https://arxiv.org/abs/2404.09636 for what seems to be the closest related works I could find?
# Important
1. Please add an evaluation of  posterior predictive coverage (e.g. using the MMD distances often used in graph generative models, e.g. https://arxiv.org/pdf/2209.14734 , or something based on the particular models chosen by the users) or justify why it is not required/feasible, from my readings it appears to be a standard sanity check in the ABI/SBI community to draw data from the inferred parameters and compare with the data distribution?
2. add a non SBI baseline (MCMC? at least for the toy model seems tractable?) => this paper https://arxiv.org/abs/2110.06581   recommends it I think
3. add computational cost across the architectures + with the MCMC baseline to the appendix, since I think this is one of the motivators for SBI?

# Some typos/nits



Page 3, Section 2.1: "the latent parameters θ govern properties the structural and statistical properties of this graph" => “gover properties  the (..)properties”

Page 2, paragraph 2: "so that node relabeling do not change h(x)" - should be "node relabelings do not change" ?

Page 10, Section 3.1.2: "a coupling flow with spline transformations build with 6 invertible layers" - should be "built with 6 invertible layers." This same error recurs on page 13 (Section 3.2.2).

Page 13, Section 3.2.3: "recovery declines as the observation horizon increased"=> increases?

Page 13, Section 3.2.2: "Active taxa are represented by zero; present taxa are stored as relative amount" => amounts?

Table 2 (Appendix A): The column header says "Loss" but the caption clarifies it's at epoch 250; meanwhile Table 3 says "Val. loss" => is table 2 val loss or train loss?

Table 3 (Appendix B): The GCN at 30 days shows "−inf" for ℓ_γ δ => what happened there?

---

> ### Author Response · Authors · 2026-02-24
>
> Thank you for your valuable and detailed feedback. I will structure the response coresponding to the section in the Requested Changes section.
>
> ## Nice to have
> Thank your for the time you put in writing this review. The named references are relevant to our work and we will add them to the paper.
>
> ## Important
> ### Posterior predictive checks and data-dependent SBC
> Posterior predictive checks are a valuable diagnostic tool; however, since all experiments in this work operate on simulated data, they are not the ideal method to investigate the validity of the data. Instead, we employ simulation-based calibration (SBC, [Modrák et al., 2025](https://projecteuclid.org/journals/bayesian-analysis/volume-20/issue-2/Simulation-Based-Calibration-Checking-for-Bayesian-Computation--The-Choice/10.1214/23-BA1404.full)), which provides richer insight into model performance. While our previous analysis focused on calibration of the model parameters, we now extend this to data-dependent test quantities. In a way, this can be seen as advanced posterior predictive checks, evaluated repeatedly on simulated datasets and a known ground truth. Specifically, we consider four graph-theoretic summary statistics: the spectral gap, edge density, degree assortativity, and global clustering coefficient. A dedicated section in the appendix presents and discusses the recovery and calibration results for all four metrics across the twelve networks examined in the toy experiment of Section 3.1. Results are summarised in the table below:
>
> Table 1: Results for the toy example across four summary-network architectures. We report the recovery (R) and the calibration ($\ell_\gamma$) of the data-dependent test quantities spectral gap (SG), edge density (ED), degree assortativity (DA) and global clustering coefficient (GC). The value is the median value across five runs.
> | Summary Network  | R SG  | R ED  | R DA  | R GC  | $\ell_\gamma$ SG | $\ell_\gamma$ ED |$\ell_\gamma$ DA |$\ell_\gamma$ GC  |
> |-----------------:|------:|------:|------:|------:|-------:|-------:|-------:|--------:|
> | Deep Sets         | 0.93  | 1.00  | 0.82  | 1.00  | -6.18  | -Inf   | 0.11   | -87.17  |
> | GCN              | 0.56  | 0.48  | 0.19  | 0.42  | -1.47  | -1.06  | 2.92   | -0.03   |
> | Graph Transformer | 0.89  | 0.94  | 0.79  | 0.92  | -24.82 | -27.45 | 2.90   | -15.69  |
> | Se tTransformer   | 0.93  | 1.00  | 0.82  | 1.00  | -22.97 | -Inf   | 0.59   | -Inf    |
>
> For the mice interaction network case study, we will include an analysis of a real world dataset. There, we will perform posterior predictive checks, where they are a natural and meaningful diagnostic. We will report results as soon as possible.
>
> ### Comparison to MCMC baseline
> Comparing our ABI method to a baseline such as MCMC is challenging because MCMC requires a tractable likelihood. In both real-world experiments — the mice interaction network and the train scheduling problem — this requirement cannot be met, as the likelihood is intractable. Even the toy example we constructed has an intractable likelihood due to the triadic closure parameter $\lambda$. A direct comparison to MCMC or similar methods is therefore not feasible in these settings.
>
> We will include a section in the appendix, where the toy case study is reduced to the parameters $\pi_{AA}, \pi_{BB}$ and $\pi_{AB}$, by removing the triadic closure parameter $\lambda$. This makes the likelihood tractable and therefore the calculation of the problem using MCMC as a baseline feasible.
>
> For this comparison, we train a BayesFlow model using the Set Transformer as the summary network. After training, we compare the resulting posterior distributions to those obtained via MCMC (specifically, adaptive HMC implemented in Stan). The posterior means are in close agreement across both methods, with similarly sized 95% credible intervals. MCMC achieves satisfactory calibration for all parameters and the log-likelihood, while the BayesFlow model is well-calibrated for the parameters but shows greater difficulty in calibrating the log-likelihood. This discrepancy is expected, as the network is trained using a simulation-based loss rather than the likelihood-based objective that underpins MCMC. We will add more detailed results including plots into a section of the appendix.

---

> > ### Author Response · Authors · 2026-02-24
> >
> > ### Computational cost
> > We agree that reporting computational cost is valuable and thank the reviewer for the suggestion. As training times were not recorded in the original experiments, we re-ran them to obtain the relevant figures. Preliminary results are reported in the table below, showing the wall-clock training time in minutes for each network architecture across varying training data sizes. Posterior sampling from the trained networks is near-instantaneous — approximately 0.1 seconds for 1000 draws — for all architectures and is therefore not separately reported.
> >
> > Table 2: Training time for the four different summary networks in minutes for four different training dataset sizes.
> > | Dataset Size | Deep Sets | GCN | Graph Transformer | Set Transformer |
> > |-------------:|-----------------------:|-------------------:|-----------------------:|---------------------:|
> > | 3200         | 4.94                   | 3.38               | 9.04                   | 13.78                |
> > | 6400         | 10.59                  | 6.41               | 29.47                  | 50.67                |
> > | 12800        | 36.40                  | 11.90              | 58.40                  | 73.95                |
> > | 32000        | 127.88                 | 41.10              | 175.56                 | 294.58               |
> >
> > ## Some typos/nits
> > Thanks for reading the paper with such an attention to detail. We fixed the typos and inconsistencies. Here the answers to the last two points:
> > - To answer the question correspond to Table 2 in Appendix A: In the toy example, the network were trained with online training, so in each epoch the model gets a unseen dataset. Therefore, the loss value presented in the table is the training loss. Table 2 in Appendix B especially tells the validation loss, because for the Mice interaction network experiment the networks are trained offline, so on a limited number of datasets.
> > - The $\ell_\gamma$ metric can take the value $-\infty$ due to numerical issues. When calibration is severely poor, $\gamma$ approaches zero, causing $\ell_\gamma$ to diverge to $-\infty$. In the tables, we report the median across five independent runs to reduce sensitivity to isolated $-\infty$ values. A reported value of $-\infty$ therefore indicates that more than half of the runs exhibited calibration poor enough to trigger this degeneracy.

---

### Author Response · Authors · 2026-03-02
**Revised version online**

Dear reviewers,
we have uploaded a revised version of our paper incorporating all requested and promised changes.

---

### Decision · Action_Editor_ZzCH · 2026-04-20

**Recommendation:** Accept as is

**Audience:**

Yes

**Audience Explanation:**

The paper sits at the intersection of amortized Bayesian inference and graph representation learning—both active areas for the TMLR audience. It should be useful for SBI/ABI practitioners looking for guidance on encoder choices for graph-structured data, as well as for GNN researchers interested in a clean benchmark setup with proper Bayesian evaluation. It’s also relevant for applied Bayesian modelling in areas with intractable likelihoods (like ecology, logistics, or epidemiology).

**Claims And Evidence:**

Yes

**Claims Explanation:**

The claims are empirical and mainly focused on comparing graph-encoder architectures within ABI, and the revision backs them up well. The authors added an MCMC baseline on a simplified version of the toy problem, included a real-data analysis with posterior predictive checks for the mice case study, and ran data-dependent SBC on several graph-based test quantities. They also show scaling curves across different simulation budgets and report training-time measurements.

In cases where standard baselines aren’t feasible (e.g., due to intractable likelihoods or because PFNs address a different inference target), their explanations are reasonable. Overall, all three reviewers agree that the evidence now aligns with the scope of the claims.